# A process model account of the role of dopamine in intertemporal choice

Alexander Soutschek[1]*, Philippe N Tobler[2,3]

[1]Department of Psychology, Ludwig Maximilian University Munich, Munich, Germany; [2]Zurich Center for Neuroeconomics, Department of Economics, University of Zurich, Zürich, Switzerland; [3]Neuroscience Center Zurich, University of Zurich, Swiss Federal Institute of Technology Zurich, Zurich, Switzerland

**Abstract** Theoretical accounts disagree on the role of dopamine in intertemporal choice and assume that dopamine either promotes delay of gratification by increasing the preference for larger rewards or that dopamine reduces patience by enhancing the sensitivity to waiting costs. Here, we reconcile these conflicting accounts by providing empirical support for a novel process model according to which dopamine contributes to two dissociable components of the decision process, evidence accumulation and starting bias. We re-analyzed a previously published data set where intertemporal decisions were made either under the D2 antagonist amisulpride or under placebo by fitting a hierarchical drift diffusion model that distinguishes between dopaminergic effects on the speed of evidence accumulation and the starting point of the accumulation process. Blocking dopaminergic neurotransmission not only strengthened the sensitivity to whether a reward is perceived as worth the delay costs during evidence accumulation (drift rate) but also attenuated the impact of waiting costs on the starting point of the evidence accumulation process (bias). In contrast, re-analyzing data from a D1 agonist study provided no evidence for a causal involvement of D1R activation in intertemporal choices. Taken together, our findings support a novel, process-based account of the role of dopamine for cost-benefit decision making, highlight the potential benefits of process-informed analyses, and advance our understanding of dopaminergic contributions to decision making.

*For correspondence:
alexander.soutschek@psy.lmu.de

**Competing interest:** The authors declare that no competing interests exist.

## Editor's evaluation

This important study reanalyzes a prior dataset testing effects of D2 antagonism on choices in a delay discounting task. While the prior report using standard analysis, showed no effects, the current study used a DDM to examine more carefully possible contrasting effects on different subcomponents of the decision process. This approach revealed convincing evidence of contrasting effects with D2 blockade increasing the effect of reward size differences to favor selection of the larger, later reward, while also shifting the bias toward selection of the small immediate reward. The authors speculate that these opposing effects explain the variability in effects across studies, since they mean that effects would depend on which of these factors is more important in a particular design.

## Introduction

Many decisions require trading-off potential benefits (rewards) against the costs of actions, such as the time one has to wait for reward to occur (*Soutschek and Tobler, 2018*). The neurotransmitter dopamine is thought to play a central role in such cost-benefit trade-offs by increasing the tolerance for action costs in order to maximize net subjective benefits (*Beeler, 2012*; *Robbins and Everitt,*

*1992*; *Salamone and Correa, 2012*; *Schultz, 2015*). Tonic dopaminergic activity was hypothesized to implement a 'cost control' which moderates whether a reward or goal is considered to be worth its costs (*Beeler and Mourra, 2018*). Prominent accounts of dopaminergic functioning thus predict that dopamine should strengthen the preference for costly larger-later (LL) over less costly smaller-sooner (SS) rewards. However, empirical studies modulating dopaminergic neurotransmission during intertemporal decision making provided inconsistent evidence for these hypotheses (for a review, see *Webber et al., 2021*). Blocking dopaminergic activation even seems to increase rather than to reduce the preference for delayed outcomes (*Arrondo et al., 2015*; *Soutschek et al., 2017*; *Wagner et al., 2020*; *Weber et al., 2016*), in apparent contrast to accounts proposing that lower dopaminergic activity should decrease the attractiveness of costly rewards (*Beeler and Mourra, 2018*; *Robbins and Everitt, 1992*; *Salamone and Correa, 2012*). Thus, the link between dopamine and cost-benefit weighting in intertemporal choice remains elusive. Yet, a plausible account of how dopamine affects cost-benefit weighting is important given that deficits in delay of gratification belong to the core symptoms of several psychiatric disorders and that dopaminergic medication plays a central role in the treatment of these and other disorders (*Hasler, 2012*; *MacKillop et al., 2011*).

To account for the conflicting findings on the role of dopamine in intertemporal choice, recent proximity accounts hypothesized that dopamine – in addition to strengthening the pursuit of valuable goals – also increases the preference for proximate over distant rewards (first formulated by *Westbrook and Frank, 2018*; see also *Soutschek et al., 2022*). While proximity and action costs often correlate negatively (as cost-free immediate rewards are typically more proximate than costly delayed rewards), they can conceptually be distinguished: perceived costs depend on an individual's internal state (e.g., available resources to wait for future rewards), whereas proximity is determined by situational factors like familiarity or concreteness (*Westbrook and Frank, 2018*). The hypothesis that dopamine increases the proximity advantage of sooner over later rewards is consistent with the observed stronger preference for LL options after D2R blockade, which could not be explained by standard accounts of the role of dopamine in cost-benefit decisions (*Beeler and Mourra, 2018*; *Salamone and Correa, 2012*).

Still, the question remains as to how the proximity account can be reconciled with the large body of evidence for a motivating role of dopamine in other domains than intertemporal choice (*Webber et al., 2021*). We recently suggested that both accounts may be unified within the framework of computational process models like the drift diffusion model (DDM) (*Soutschek et al., 2022*). DDMs assume that decision makers accumulate evidence for two reward options until a decision boundary is reached. The dopamine-mediated cost control may be implemented via dopaminergic effects on the evaluation of reward magnitudes and delay costs during the evidence accumulation process (drift rate), while a proximity advantage for sooner over delayed rewards may shift the starting bias toward the decision boundary for sooner rewards (*Soutschek et al., 2022*; *Westbrook and Frank, 2018*). Such proximity effects on the starting bias could reflect an automatic bias toward immediate rewards as posited by dual process models of intertemporal choice (*Figner et al., 2010*; *McClure et al., 2004*), whereas the influence of reward and delay on the drift rate involves more controlled and attention-demanding weighting of costs and benefits. Combining these two, in their consequences on overt choices partially opposing, but independent, effects of dopamine in a unified and tractable account could reconcile conflicting findings. In turn, such a process account might provide a knowledge basis to advance our understanding of the neurochemical basis of the decision-making deficits in clinical disorders and improve the effectiveness of pharmaceutical interventions.

Here, we tested central assumptions of the proposed account by re-analyzing the data from two previous studies that investigated how the dopamine D2 receptor antagonist amisulpride and the D1 agonist PF-06412562 impact cost-benefit weighting in intertemporal choice (*Soutschek et al., 2017*; *Soutschek et al., 2020a*). D1Rs are prevalent in the direct 'Go' pathway and facilitate action selection via mediating the impact of phasic bursts elicited by high above-average rewards (*Evers et al., 2017*; *Kirschner et al., 2020*). D2Rs, in contrast, dominate the indirect 'Nogo' pathway (which suppresses action) and are more sensitive to small concentration differences in tonic dopamine levels (*Missale et al., 1998*), which is thought to encode the background, average reward rate (*Kirschner et al., 2020*; *Volkow and Baler, 2015*; *Westbrook and Frank, 2018*; *Westbrook et al., 2020*). Comparing the influences of the two compounds on the choice process during intertemporal decisions allowed us to test the hypothesized dissociable roles of D1Rs and D2Rs for decision making. Previously reported

analyses of these data had shown no influence of D2R blockade or D1R stimulation on the mean preferences for LL over SS options (*Soutschek et al., 2017*; *Soutschek et al., 2020a*). However, they had not asked whether the pharmacological agents moderate the influences of reward magnitudes and delay costs on subcomponents of the decision process within the framework of a DDM. We re-analyzed the data sets with hierarchical Bayesian drift diffusion modeling to test central assumptions of the proposed account on dopamine's role in cost-benefit weighting. First, if D2R activation implements a cost threshold moderating the evaluation of whether a reward is worth the action costs, then blocking D2R activation with amisulpride should increase the influence of reward magnitude on the speed of evidence accumulation, with costly small rewards becoming less acceptable than under placebo. Second, if D2R-mediated tonic dopaminergic activity also moderates the impact of proximity on choices (which affects the starting bias rather than the speed of the evidence accumulation process), D2R blockade should attenuate the effects of waiting costs on the starting bias. Third, we expected D1R stimulation to modulate the sensitivity to rewards during evidence accumulation (via increasing activity in the direct 'Go' pathway), without affecting proximity costs which were related to tonic rather than phasic dopaminergic activity (*Westbrook and Frank, 2018*).

## Results

To disentangle how dopamine contributes to distinct subcomponents of the choice process, we re-analyzed a previously published data set where 56 participants had performed an intertemporal choice task under the D2 antagonist amisulpride (400 mg) and placebo in two separate sessions (*Soutschek et al., 2017*; *Figure 1*). First, we assessed amisulpride effects on intertemporal choices with conventional model-based and model-free analyses, as they are employed by other pharmacological studies on cost-benefit weighting. Hyperbolic discounting of future rewards was not significantly different under amisulpride (mean log-k=–2.07) compared with placebo (mean log-k=–2.19), Bayesian t-test, $HDI_{mean} = 0.21$, $HDI_{95\%} = [–0.28; 0.70]$, and there were also no drug effects on choice consistency (inverse temperature), $HDI_{mean} = –0.28$, $HDI_{95\%} = [–0.71; 0.13]$. Model-free Bayesian mixed generalized linear models (MGLMs) revealed a stronger preference for LL over SS options with increasing differences in reward magnitudes, $HDI_{mean} = 6.32$, $HDI_{95\%} = [5.03; 7.83]$, and with decreasing differences in delay of reward delivery, $HDI_{mean} = –1.27$, $HDI_{95\%} = [–1.87; –0.60]$. The impact of delays on choices was significantly reduced under amisulpride compared with placebo, $HDI_{mean} = 0.75$, $HDI_{95\%} = [0.02; 1.67]$ (*Figure 1C/D* and *Table 1*). When we explored whether dopaminergic effects changed over the course of the experiment, we observed a significant main effect of trial number (more LL choices over time), $HDI_{mean} = 0.58$, $HDI_{95\%} = [0.19; 0.99]$. However, this effect was unaffected by the pharmacological manipulation, $HDI_{mean} = –0.06$, $HDI_{95\%} = [–0.61; 0.48]$. We also re-computed the MGLM reported above on log-transformed decision times, adding predictors for choice (SS vs. LL option) and Magnitude$_{sum}$ (combined magnitudes of SS and LL rewards). Participants made faster decisions the higher the sum of the two rewards, $HDI_{mean} = –0.12$, $HDI_{95\%} = [–0.18; –0.06]$, however we observed no significant drug effects on decision times. Thus, based on these conventional analyses one would conclude that reduction of D2R neurotransmission lowers the sensitivity to delay costs, which on the one hand agrees with one line of previous findings (*Arrondo et al., 2015*; *Wagner et al., 2020*; *Weber et al., 2016*). On the other hand, this result seems to contradict the widely held assumption that dopamine increases the preference for costly over cost-free outcomes (*Beeler and Mourra, 2018*; *Webber et al., 2021*; *Westbrook et al., 2020*), because according to this view lower dopaminergic activity should increase, rather than decrease, the impact of waiting costs on LL choices. However, analyses that consider only the observed choices do not allow disentangling dopaminergic influences on distinct subcomponents of the choice process.

DDMs paint a fuller picture of the decision process than pure choice data by integrating information from observed choices and decision times. DDMs assume that agents accumulate evidence for the choice options (captured by the drift parameter v) from a starting point $\zeta$ until the accumulated evidence reaches a decision threshold (boundary parameter a; *Figure 1E*). Following previous procedures analyzing intertemporal choices with DDMs (*Amasino et al., 2019*), we assumed that the drift rate $v$ integrates reward magnitudes and delays of choice options via attribute-wise comparisons (DDM-1). In addition, we also allowed the starting bias to vary as a function of differences in delay costs, in line with recent proximity accounts of dopamine (*Westbrook and Frank, 2018*).

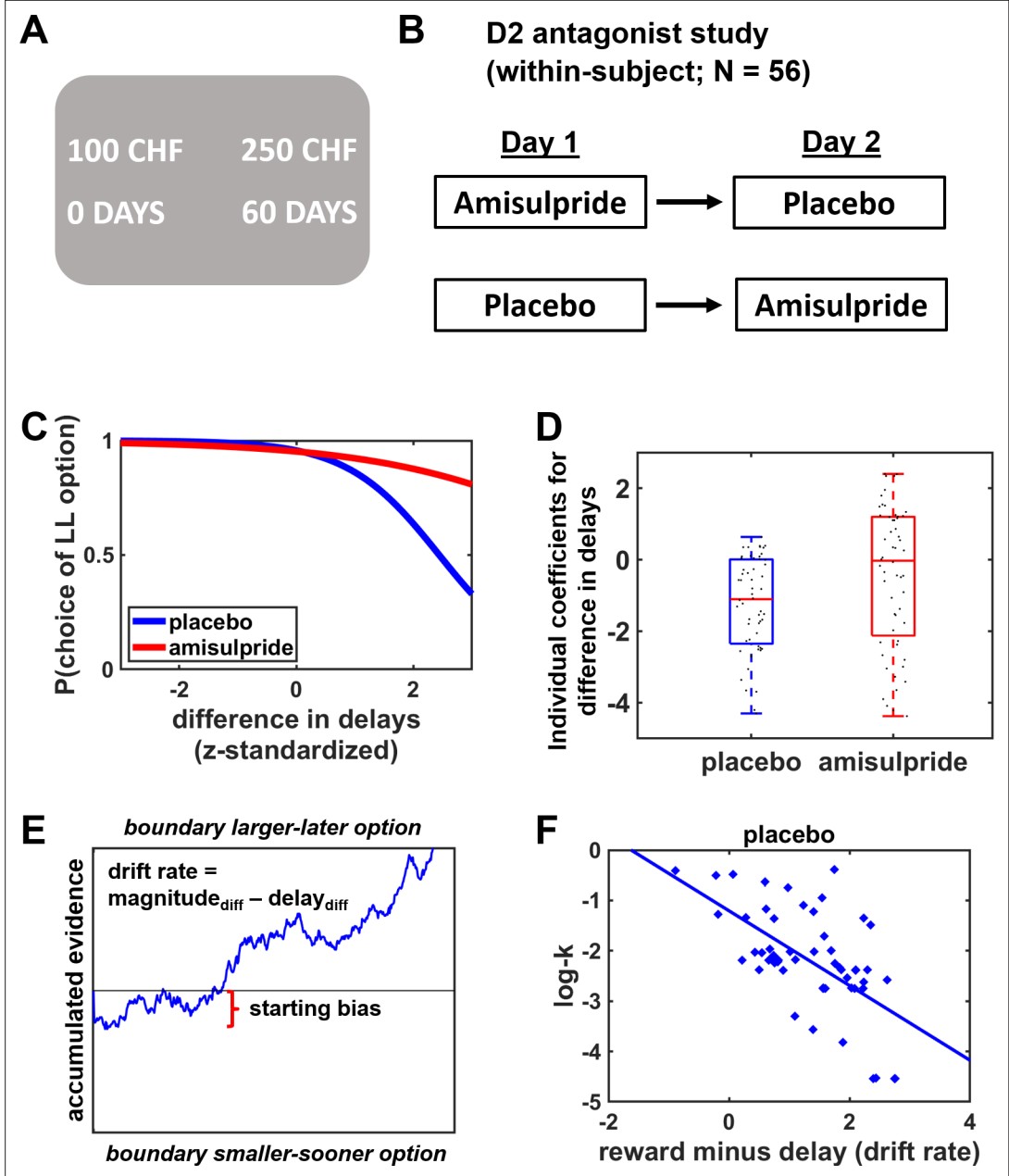

**Figure 1.** Task design and experimental procedures. (**A**) Participants made choices between alternatives that provided smaller-sooner rewards (e.g., 100 Swiss francs in 0 day) or larger-later rewards (e.g., 250 Swiss francs in 60 days). (**B**) In a double-blind crossover design, participants performed the intertemporal decision task after administration of the D2 antagonist amisulpride or placebo on two separate days. (**C**) Model-free Bayesian analyses revealed weaker influences of delay costs on decision making under amisulpride compared with placebo, consistent with previous findings that D2R antagonism strengthens the preference for delayed rewards. (**D**) Individual coefficients for the impact of delay on choices in the amisulpride and placebo conditions. (**E**) Illustration of the choice process in the framework of a drift-diffusion model. After a non-decision time $\tau$ (not shown here), evidence is accumulated from a starting point $\zeta$ with the weighted difference between benefits and action costs determining the speed of the accumulation process (drift rate v) toward the boundaries for the larger-later or smaller-sooner option. (**F**) Delay discounting under placebo (log-k; dots correspond to individual participants, with more negative values indicating weaker delay discounting) decreased with the difference in weights assigned to rewards and delay costs during evidence accumulation, replicating previous findings (*Amasino et al., 2019*).

A sanity check revealed that larger differences between the reward magnitudes of the LL and SS options bias evidence accumulation toward the LL option, $HDI_{mean}$ = 2.41, $HDI_{95\%}$ = [1.93; 2.95], whereas larger differences in delays bias accumulation in favor of the SS option, $HDI_{mean}$ = −1.13, $HDI_{95\%}$ = [−1.53; −0.78]. Moreover, we assessed the relationship between the difference in DDM parameters

**Table 1.** Results of Bayesian generalized linear model regressing binary choices (larger-later [LL] vs. smaller-sooner [SS] option) in the amisulpride study on predictors for Drug, Magnitude$_{diff}$, Delay$_{diff}$, and the interaction terms.

Standard errors of the mean of the posterior distributions are in brackets.

| Predictor | Mean | 2.5% | 97.5% |
|---|---|---|---|
| Intercept | 3.10 (0.64) | 1.91 | 4.43 |
| Drug | –0.10 (0.53) | –1.12 | 0.96 |
| Delay$_{diff}$ | –1.27 (0.32) | –1.87 | –0.60 |
| Magnitude$_{diff}$ | 6.32 (0.72) | 5.03 | 7.83 |
| Drug×Delay$_{diff}$ | 0.75 (0.41) | 0.02 | 1.67 |
| Drug×Magnitude$_{diff}$ | 0.25 (0.71) | –1.05 | 1.77 |
| Delay$_{diff}$ ×Magnitude$_{diff}$ | 0.16 (0.43) | –0.61 | 1.10 |
| Drug×Delay$_{diff}$×Magnitude$_{diff}$ | 0.73 (0.60) | –0.30 | 2.07 |

(reward magnitude – delay) and hyperbolic discount parameters log-k as purely choice-based indicator of impulsiveness. Replicating previous findings, we found that across individuals the weights relate to delay discounting, r=–0.61, p<0.001 (*Amasino et al., 2019*; *Figure 1F*), such that individuals weighting reward magnitudes more strongly over delays make more patient choices. Thus, our model parameters capture essential subprocesses of intertemporal decision making.

Next, we tested the impact of our dopaminergic manipulation on evidence accumulation: D2R blockade strengthened the impact of differences in reward magnitude on evidence accumulation, Drug×Magnitude$_{diff}$: HDI$_{mean}$ = 0.81, HDI$_{95\%}$ = [0.04; 1.71], while the contribution of differences in delay costs remained unchanged, Drug×Delay$_{diff}$: HDI$_{mean}$ = –0.30, HDI$_{95\%}$ = [–0.85; 0.20] (*Figure 2A–F* and *Table 2*). The drug-induced increase in sensitivity to variation in reward magnitude suggests that low rewards are considered less valuable under amisulpride compared with placebo (*Figure 2C*). This finding is consistent with the cost control hypothesis (*Beeler and Mourra, 2018*) according to which low dopamine levels reduce the attractiveness of smaller, below-average rewards.

When we assessed dopaminergic effects on the starting bias, we observed that under placebo increasing differences in delay shifted the starting point toward the SS option, HDI$_{mean}$ = 0.81, HDI$_{95\%}$ = [0.04; 1.71], suggesting that the bias parameter is closer to the proximate (SS) option the stronger the proximity advantage of the SS over the LL option. Amisulpride shifted the starting bias toward the SS option for smaller differences in delay, main effect of Drug: HDI$_{mean}$ = –0.04, HDI$_{95\%}$ = [–0.08; -0.001], but also attenuated the impact of delay, Drug×Delay$_{diff}$: HDI$_{mean}$ = 0.02, HDI$_{95\%}$ = [0.001; 0.04]. Thus, dopamine appears to moderate the impact of temporal proximity on the starting bias (*Figure 2G–K*), providing support for recent proximity accounts of dopamine (*Soutschek et al., 2022*; *Westbrook and Frank, 2018*). Moreover, compared to the model-free analysis, our process model (which uses not only binary choice but also response time data) provides a fuller picture of the subcomponents of the choice process affected by the dopaminergic manipulation.

Next, we investigated the relation between the drug effects on the drift rate and on the starting bias. We found no evidence that the two effects correlated, r=0.07, p=0.60, suggesting that amisulpride effects on these subprocesses were largely independent of each other. Control analyses revealed no effects of amisulpride on non-decision times, HDI$_{mean}$ = –0.10, HDI$_{95\%}$ = [–0.24; 0.03], or the decision threshold, HDI$_{mean}$ = 0.17, HDI$_{95\%}$ = [–0.11; 0.46]. Thus, the results of DDM-1 suggest that dopamine moderates the influence of choice attributes on both the speed of evidence accumulation and on the starting bias, consistent with recent accounts (*Soutschek et al., 2022*; *Westbrook and Frank, 2018*) of dopamine's role in cost-benefit weighting.

To test the robustness of our DDM findings, we computed further DDMs where we either removed the impact of Delay$_{diff}$ on the starting bias (DDM-2) or the impact of Magnitude$_{diff}$ and Delay$_{diff}$ on the drift rate (DDM-3). In a further model (DDM-4), we explored whether the starting bias is affected by the overall proximity of the options (sum of delays, Delay$_{sum}$) rather than the difference in proximity (Delay$_{diff}$; see *Table 3* for an overview over the parameters included in the various models). Importantly, our original DDM-1 (DIC = 9478) explained the data better than DDM-2 (DIC = 9481), DDM-3

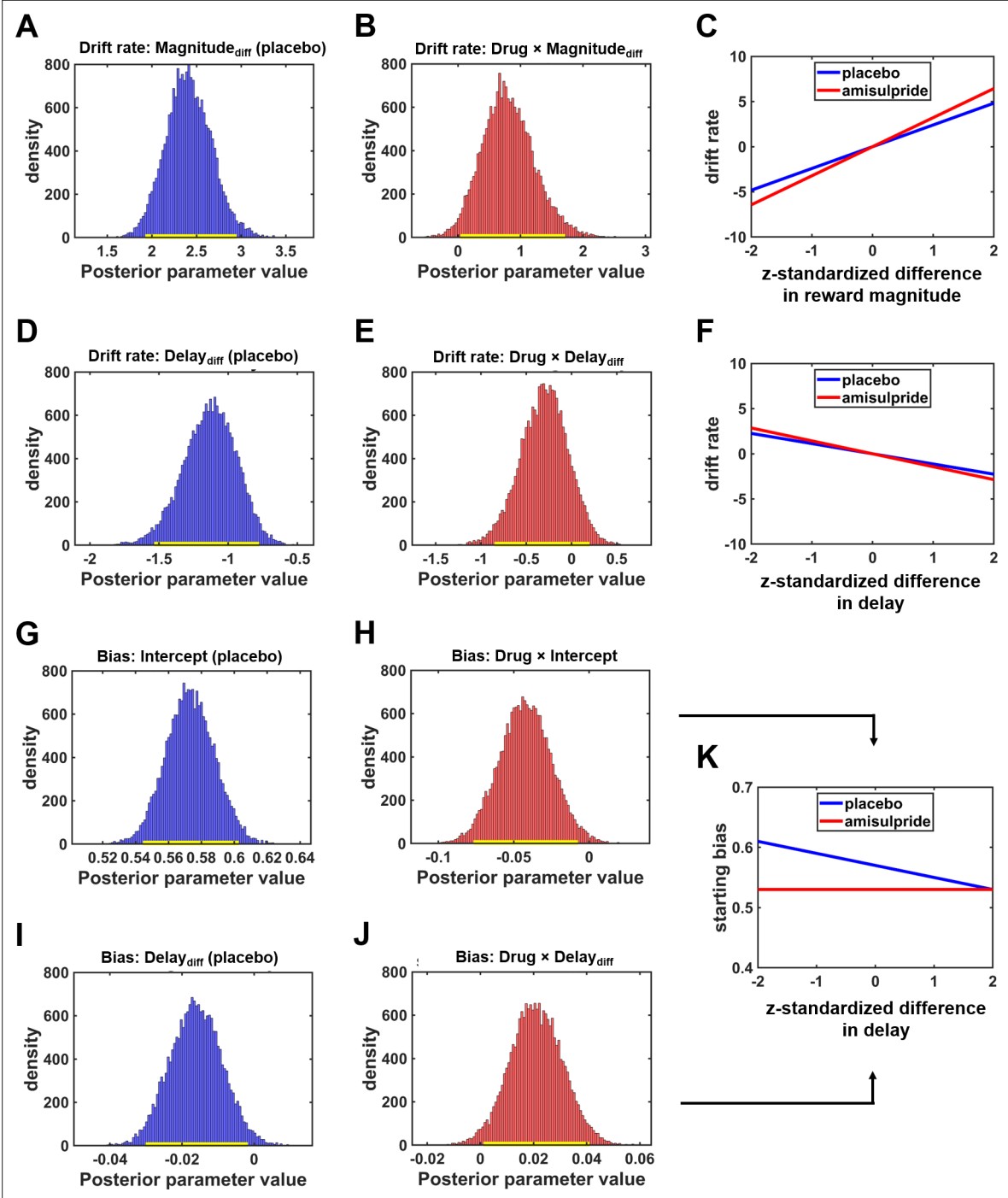

**Figure 2.** D2R blockade affects multiple components of the intertemporal decision process. (**A**) Larger differences in reward magnitude between the larger-later (LL) and smaller-sooner (SS) option increased drift rates, speeding up evidence accumulation toward LL options under placebo. (**B**) The impact of differences in reward magnitude was significantly stronger under amisulpride than under placebo. (**C**) Drug-dependent impact of differences in reward magnitude on the drift rate. Because the sensitivity to differences in reward magnitude was stronger under amisulpride than under placebo (steeper slope), D2R blockade sped up evidence accumulation toward the boundary for LL choices if differences in reward magnitude between the LL and SS options were large. In contrast, if the difference in reward magnitude was small, the drift rate was more negative under amisulpride compared with placebo, speeding up evidence accumulation toward the SS option. (**D**) Larger differences between delay of reward promoted evidence accumulation toward the (negative) boundary for SS choices under placebo, (**E, F**) but the impact of delay was not significantly altered by amisulpride. (**G**) The starting point of the accumulation processes was closer to the boundary for LL than SS choices under placebo, (**H**) and this starting bias toward the LL option was significantly reduced by amisulpride. (**I**) For larger differences in waiting costs, the starting point of the evidence accumulation process was increasingly shifted toward the SS option under placebo. (**J**) This impact of delay costs on the starting bias was significantly reduced under

*Figure 2 continued on next page*

*Figure 2 continued*

amisulpride. As illustrated in (**K**), reducing dopaminergic action on D2R with amisulpride shifted the starting bias toward the boundary for the SS option predominantly if no option possessed a clear proximity advantage (small difference between delays). In A, B, D, E, G, H, I, and J, yellow bars close to x-axis indicate 95% HDIs.

(DIC = 10,224), or DDM-4 (DIC = 9492; *Figure 3A*). Nevertheless, amisulpride moderated the impact of Magnitude$_{diff}$ on the drift rate also in DDM-2, HDI$_{mean}$ = 0.86, HDI$_{95\%}$ = [0.18; 1.64], and DDM-4, HDI$_{mean}$ = 0.83, HDI$_{95\%}$ = [0.04; 1.75], and amisulpride also lowered the impact of Delay$_{diff}$ on the starting bias in DDM-3, HDI$_{mean}$ = –0.02, HDI$_{95\%}$ = [–0.04; –0.001]. Thus, the dopaminergic effects on these subcomponents of the choice process are robust to the exact specification of the DDM.

We compared the winning account also with alternative process models of intertemporal choice. While in DDM-1 the drift rate depends on separate comparisons between choice attributes, one might alternatively assume that they compare the discounted subjective reward values of both options (*Wagner et al., 2020*), as given by the hyperbolic discount functions. However, a DDM where the drift rate was modeled as the difference between the hyperbolically discounted reward values (with the discount factor as free parameter; DDM-5) showed a worse model fit (DIC = 10,720) than DDM-1. This replicates previous findings according to which intertemporal choices can better be explained by attribute-wise than by option-wise comparison strategies (*Amasino et al., 2019*; *Dai and Busemeyer, 2014*; *Reeck et al., 2017*).

Next, we investigated an alternative to the proposal that differences in delay affect the starting bias via proximity effects. Specifically, we tested whether evidence for delay costs are accumulated earlier than for reward magnitude (relative-starting-time-(rs)DDM; *Amasino et al., 2019*; *Lombardi and Hare, 2021*). From the perspective of rsDDMs, evidence accumulation for delays would start after a shorter non-decision time than for rewards, which is expressed by the variable $\tau_{diff}$ (if $\tau_{diff} > 0$, non-decision time is shorter for delays than rewards, and vice versa if $\tau_{diff} < 0$). However, also this rsDDM (DDM-6) explained the data less well (DIC = 9,548) than DDM-1. Thus, DDM-1 explains the current data better than alternative DDMs.

The currently used dose of amisulpride (400 mg) is thought to have predominantly postsynaptic effects on D2Rs, while lower doses (50–300 mg) might show presynaptic rather than postsynaptic effects (*Schoemaker et al., 1997*). Given that we used the same dose in all participants, one might argue that we may have studied presynaptic effects in individuals with relatively high body mass (which

**Table 2.** Results of hierarchical DDM-1 for the amisulpride data.
SEM: standard errors of the mean of the posterior distributions.

| Parameter | Regressor | Mean (SEM) | 2.5% | 97.5% |
|---|---|---|---|---|
| Drift rate | Delay$_{diff}$ | –1.13 (0.19) | –1.53 | –0.78 |
| | Magnitude$_{diff}$ | 2.41 (0.26) | 1.93 | 2.95 |
| | Drug×Delay$_{diff}$ | –0.30 (0.27) | –0.85 | 0.20 |
| | Drug×Magnitude$_{diff}$ | 0.81 (0.42) | 0.04 | 1.71 |
| | v$_{max}$ | 0.80 (0.05) | 0.70 | 0.90 |
| | Drug×v$_{max}$ | –0.00 (0.06) | –0.12 | 0.12 |
| Decision threshold | Placebo | 3.96 (0.15) | 3.67 | 4.26 |
| | Drug | 0.17 (0.15) | –0.11 | 0.46 |
| Starting bias | Placebo | 0.57 (0.01) | 0.54 | 0.60 |
| | Drug | –0.04 (0.02) | –0.08 | –0.001 |
| | Delay$_{diff}$ | –0.02 (0.01) | –0.03 | –0.002 |
| | Drug×Delay$_{diff}$ | 0.02 (0.01) | 0.001 | 0.04 |
| Non-decision time | Placebo | 1.48 (0.06) | 1.36 | 1.60 |
| | Drug | –0.10 (0.07) | –0.24 | 0.03 |

**Table 3.** Overview over drift diffusion model (DDM) parameters included in the DDMs.
Note that we modeled drug effects for all parameters included in the DDMs.

| Parameter | | DDM-1 | DDM-2 | DDM-3 | DDM-4 | DDM-5 | DDM-6 |
|---|---|---|---|---|---|---|---|
| Drift rate | Delay$_{diff}$ | ✓ | ✓ | | ✓ | ✓ | ✓ |
| | Magnitude$_{diff}$ | ✓ | ✓ | | ✓ | | ✓ |
| | $v_{max}$ | ✓ | ✓ | ✓ | ✓ | ✓ | ✓ |
| Decision threshold | Intercept | ✓ | ✓ | ✓ | ✓ | ✓ | ✓ |
| Starting bias | Intercept | ✓ | | ✓ | ✓ | ✓ | ✓ |
| | Delay$_{diff}$ | ✓ | | ✓ | | ✓ | |
| | Delay$_{sum}$ | | | | ✓ | | |
| Non-decision time | Intercept | ✓ | ✓ | ✓ | ✓ | ✓ | ✓ |
| | $\tau_{diff}$ | | | | | | ✓ |

lowers the effective dose). However, we observed no evidence that individual random coefficients for the drug effects on the drift rate or on the starting bias correlated with body weight, all r<0.22, all p>0.10. There were also no significant correlations between DDM parameters and performance in the digit span backward task as proxy for baseline dopamine synthesis capacity (*Cools et al., 2008*), all r<0.17, all p>0.22. There was thus no evidence that pharmacological effects on intertemporal choices depended on body weight as proxy of effective dose or working memory performance as proxy for baseline dopaminergic activity.

As further check of the explanatory adequacy of DDM-1, we performed posterior predictive checks and parameter recovery analyses. Plotting the observed RTs (split into quintiles according to Magnitude$_{diff}$ and Delay$_{diff}$) against the simulated RTs based on the parameter estimates from the different DDMs suggests that the DDMs provide reasonable accounts of the observed data both on the group and the individual level, at least for DDMs 1–3 (*Figure 3B/C* and *Figure 4*). Moreover, the squared differences between observed and simulated RTs were smaller for DDM-1 (0.83) than for alternative DDMs (DDM-2: 0.85; DDM-3: 0.98; DDM-4: 1.05, DDM-5: 0.89; DDM-6: 1.63). To assess parameter recovery, we re-computed DDM-1 on 10 simulated data sets based on the original DDM-1 parameter estimates. All group-level parameters from the simulated data were within the 95% HDI of the original parameter estimates, except for the non-decision time $\tau$ (which suggests that our model tends to overestimate the duration of decision-unrelated processes). Nevertheless, all parameters determining

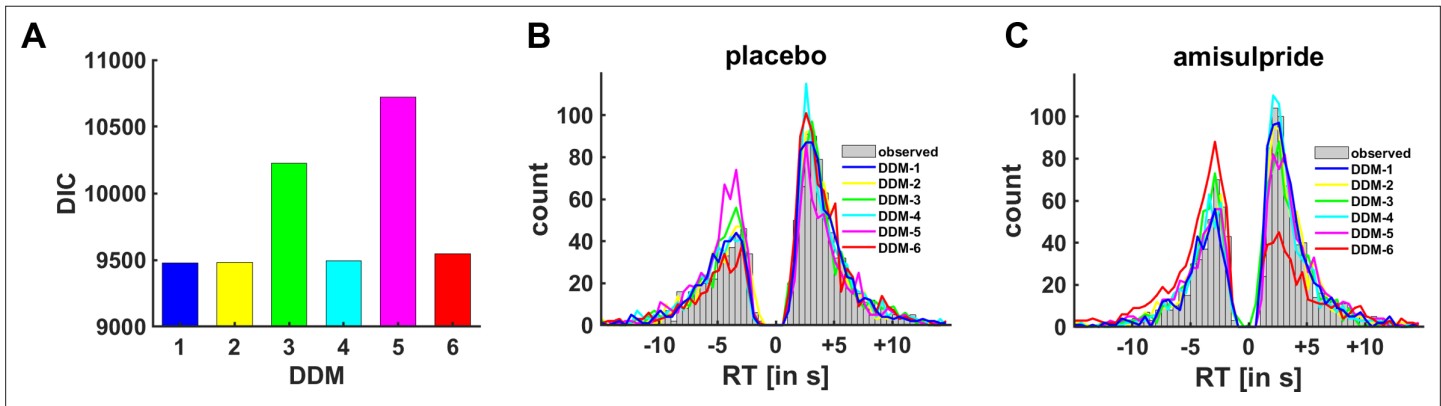

**Figure 3.** Model comparison.
(**A**) The deviance information criterion (DIC; lower numbers correspond to better fit) suggests that DDM-1 explained the data slightly better than DDM-2, DDM-4, and DDM-6 and clearly outperformed DDM-3 and DDM-5. (**B, C**) Posterior predictive checks on the group level (collapsed across all participants), separately for (**B**) placebo and (**C**) amisulpride. Particularly DDM-1, DDM-2, and DDM-3 described the empirically observed data well, whereas decisions simulated based on DDMs 4–6 more strongly deviated from observed behavior.

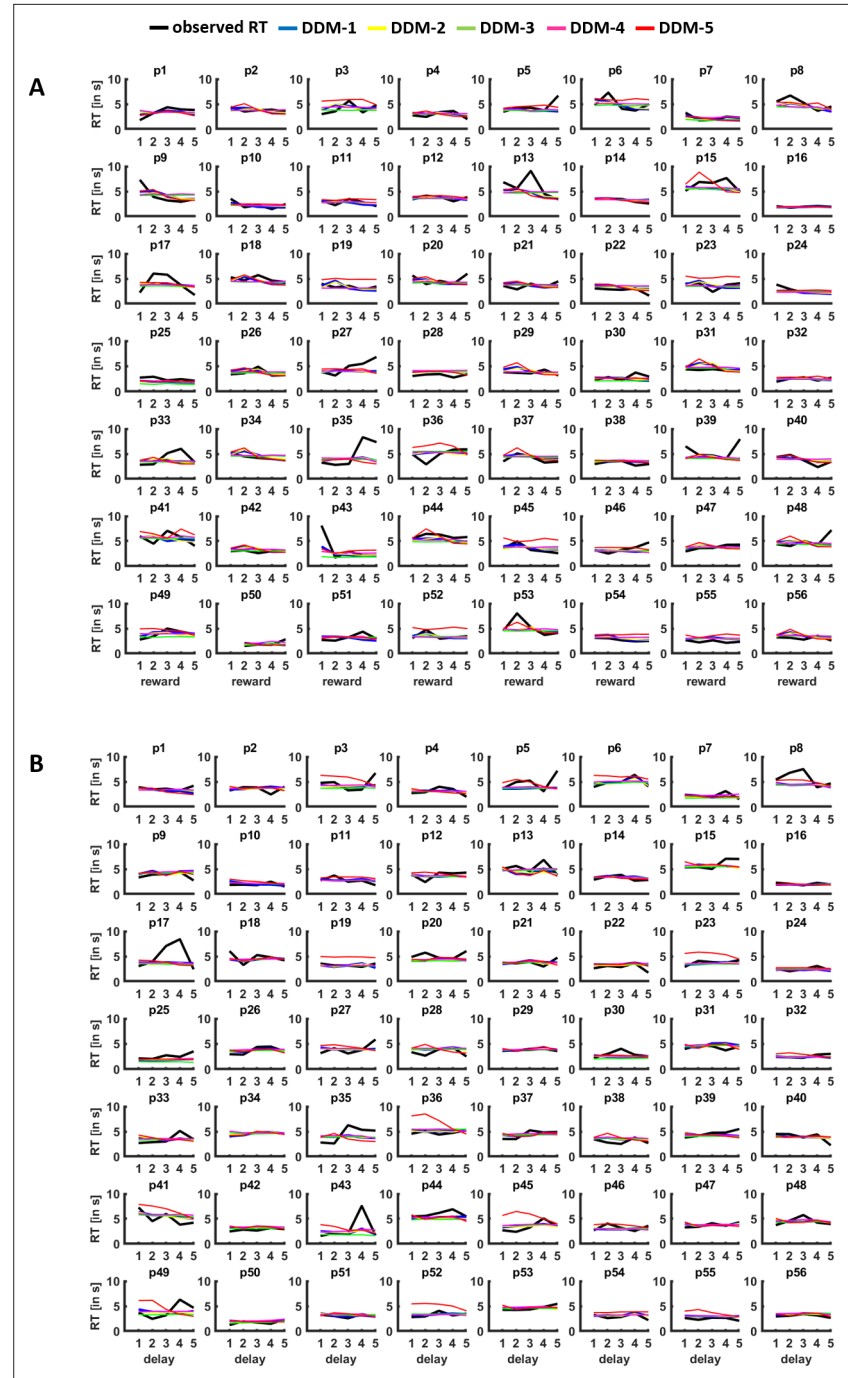

**Figure 4.** Posterior predictive checks. For each individual participant (**p1–p56**), observed RTs (in black) are plotted against the RTs simulated based on the parameters for drift diffusion model (DDM) 1–6, separately for differences in (**A**) reward magnitude and (**B**) delay (quintiles). The plots suggest that the DDMs provide reasonable accounts of the observed RTs.

the outcome of the decision process (i.e., the choice made) as well as the dopaminergic effects on the parameters could reliably be recovered by DDM-1.

To assess the receptor specificity of our findings, we conducted the same analyses on the data from a study (published previously in *Soutschek et al., 2020a*) testing the impact of three doses of a D1 agonist (6, 15, 30 mg) relative to placebo on intertemporal choices (between-subject design). In the intertemporal choice task used in this experiment, the SS reward was always immediately available

**Table 4.** Results of the best-fitting DDM-1 for the D1 agonist experiment.
Standard errors of the mean of the posterior distributions are in brackets.

| Parameter | Regressor | Mean | 2.5% | 97.5% |
|---|---|---|---|---|
| Drift rate: $\text{Delay}_{diff}$ | Placebo (0 mg) | –0.42 | –0.74 | –0.20 |
| | 6 mg vs. 0 mg | –0.02 | –0.35 | 0.28 |
| | 15 mg vs. 0 mg | –0.15 | –0.52 | 0.19 |
| | 30 mg vs. 0 mg | –0.12 | –0.50 | 0.14 |
| Drift rate: $\text{Magnitude}_{diff}$ | Placebo (0 mg) | 0.86 | 0.49 | 1.31 |
| | 6 mg vs. 0 mg | 0.17 | –0.26 | 0.63 |
| | 15 mg vs. 0 mg | 0.11 | –0.53 | 0.73 |
| | 30 mg vs. 0 mg | 0.23 | –0.23 | 0.83 |
| Drift rate: $v_{max}$ | Placebo (0 mg) | 1.72 | 1.15 | 2.66 |
| | 6 mg vs. 0 mg | –0.36 | –1.37 | 0.60 |
| | 15 mg vs. 0 mg | 0.02 | –1.18 | 1.35 |
| | 30 mg vs. 0 mg | –0.39 | –1.60 | 0.75 |
| Decision threshold | Placebo (0 mg) | 2.82 | 2.59 | 3.04 |
| | 6 mg vs. 0 mg | –0.13 | –0.46 | 0.20 |
| | 15 mg vs. 0 mg | –0.11 | –0.45 | 0.24 |
| | 30 mg vs. 0 mg | –0.06 | –0.39 | 0.25 |
| Starting bias: Intercept | Placebo (0 mg) | 0.59 | 0.54 | 0.64 |
| | 6 mg vs. 0 mg | –0.01 | –0.07 | 0.06 |
| | 15 mg vs. 0 mg | –0.01 | –0.08 | 0.06 |
| | 30 mg vs. 0 mg | –0.03 | –0.09 | 0.04 |
| Starting bias: $\text{Delay}_{diff}$ | Placebo (0 mg) | 0.00 | –0.01 | 0.02 |
| | 6 mg vs. 0 mg | –0.01 | –0.03 | 0.02 |
| | 15 mg vs. 0 mg | 0.01 | –0.01 | 0.04 |
| | 30 mg vs. 0 mg | –0.01 | –0.03 | 0.01 |
| Non-decision time | Placebo (0 mg) | 0.85 | 0.78 | 0.93 |
| | 6 mg vs. 0 mg | 0.03 | –0.09 | 0.15 |
| | 15 mg vs. 0 mg | –0.02 | –0.13 | 0.09 |
| | 30 mg vs. 0 mg | 0.03 | –0.06 | 0.13 |

(delay = 0), contrary to the task in the D2 experiment where the delay of the SS reward varied from 0 to 30 days. Again, the data in the D1 experiment were best explained by DDM-1 ($\text{DIC}_{DDM-1}$=19,657) compared with all other DDMs ($\text{DIC}_{DDM-2}$=20,934; $\text{DIC}_{DDM-3}$=21,710; $\text{DIC}_{DDM-5}$=21,982; $\text{DIC}_{DDM-6}$=19,660; note that DDM-4 was identical with DDM-1 for the D1 agonist study because the delay of the SS reward was 0). Neither the best-fitting nor any other model yielded significant drug effects on any drift diffusion parameter (see *Table 4* for the best-fitting model). Also model-free analyses conducted in the same way as for the D2 antagonist study revealed no significant drug effects (all $\text{HDI}_{95\%}$ included zero). There was thus no evidence for any influence of D1R stimulation on intertemporal decisions.

## Discussion

Dopamine is hypothesized to play a central role in human cost-benefit decision making, but existing empirical evidence does not conclusively support the widely shared assumption that dopamine

promotes the pursuit of high benefit-high cost options (for reviews, see *Soutschek et al., 2022*; *Webber et al., 2021*). By manipulating dopaminergic activity with the D2 antagonist amisulpride, we provide empirical evidence for a novel process model of cost-benefit weighting that reconciles conflicting views by assuming dissociable effects of dopamine on distinct subcomponents of the decision process.

D2R blockade (relative to placebo) increased the sensitivity to variation in reward magnitudes during evidence accumulation, such that only relatively large future rewards were considered to be worth the waiting cost, whereas small delayed rewards were perceived as less valuable than sooner rewards. This dopaminergic impact on the drift rate is consistent with the view that D2R-mediated tonic dopamine levels implement a cost control determining whether a reward is worth the required action costs (*Beeler and Mourra, 2018*). From this perspective, lowering D2R activity with amisulpride resulted in a stricter cost control such that only rather large delayed rewards were able to overcome D2R-mediated cortical inhibition (*Lerner and Kreitzer, 2011*). While this effect is consistent with the standard view according to which dopamine increases the preference for large costly rewards (*Robbins and Everitt, 1992*; *Salamone and Correa, 2012*; *Schultz, 2015*), the dopaminergic effects on the starting bias parameter yielded a different pattern. Here, inhibition of D2R activation reduced the impact of delay costs on the starting bias, such that for shorter delays (where the immediate reward has only a small proximity advantage) D2R inhibition shifts the bias toward the SS option. This finding represents first evidence for the hypothesis that tonic dopamine moderates the impact of proximity (e.g., more concrete vs. more abstract rewards) on cost-benefit decision making (*Soutschek et al., 2022*; *Westbrook and Frank, 2018*). Pharmacological manipulation of D1R activation, in contrast, showed no significant effects on the decision process. This provides evidence for the receptor specificity of dopamine's role in intertemporal decision making (though as caveat it is worth keeping the differences between the tasks administered in the D1 and the D2 studies in mind).

Conceptually, the assumption of proximity effects on the starting bias is consistent with dual process models of intertemporal choice assuming that individuals are (at least partially) biased toward selecting immediate over delayed rewards (*Figner et al., 2010*; *McClure et al., 2004*). This automatic favoring of immediate rewards is reflected in a shift of the starting bias and thus occurs before the evidence accumulation process, which relies on attention-demanding cost-benefit weighting (*Zhao et al., 2019*). In agreement with this notion, DDM-1 with temporal proximity-dependent bias showed better fit than DDM-5 with variable non-decision times for rewards and delays. We note that the hierarchical modeling approach allowed us to compare models on the group-level only, such that in some individuals behavior might better be explained by a different model than DDM-1. Such model comparisons on the individual level, however, were beyond the scope of the current study and might not yield robust results given the limited number of trials per individual. We also emphasize that alternative process models like the linear ballistic accumulator (LBA) model make different assumptions than DDMs, for example by positing the existence of separate option-specific accumulators rather than only one as assumed by DDMs. However, proximity effects as investigated in the current study might be incorporated in LBA models as well by varying the starting points of the accumulators as function of proximity.

A dopaminergic modulation of proximity effects provides an elegant explanation for the fact that in most D2 antagonist studies D2R reduction increased the preference for LL options (*Arrondo et al., 2015*; *Soutschek et al., 2017*; *Wagner et al., 2020*; *Weber et al., 2016*), contrary to the predictions of energization accounts (*Beeler and Mourra, 2018*; *Salamone and Correa, 2012*). Noteworthy, the dopaminergic effects on evidence accumulation and on the starting bias promote potentially different action tendencies, as the impact of amisulpride on evidence accumulation lowered the weight assigned to small future rewards, whereas the amisulpride effects on the starting bias increased the likelihood of LL options being chosen. Rather than generally biasing impulsive or patient choices, the impact of dopamine on decision making may therefore crucially depend on the rewards at stake and the associated waiting costs (*Figure 4*). In our model, lower dopamine levels strengthen the preference for high reward-high cost options predominantly in two situations. First, if differences in reward magnitude are high (e.g., choosing between your favorite meal vs. a clearly less liked dish) and, second, if the less costly option has a clear proximity advantage over the costlier one (having dinner in a restaurant close-by or a preferred restaurant on the other side of town). Conversely, if differences in both expected reward and waiting costs are small, lower dopamine may bias choices

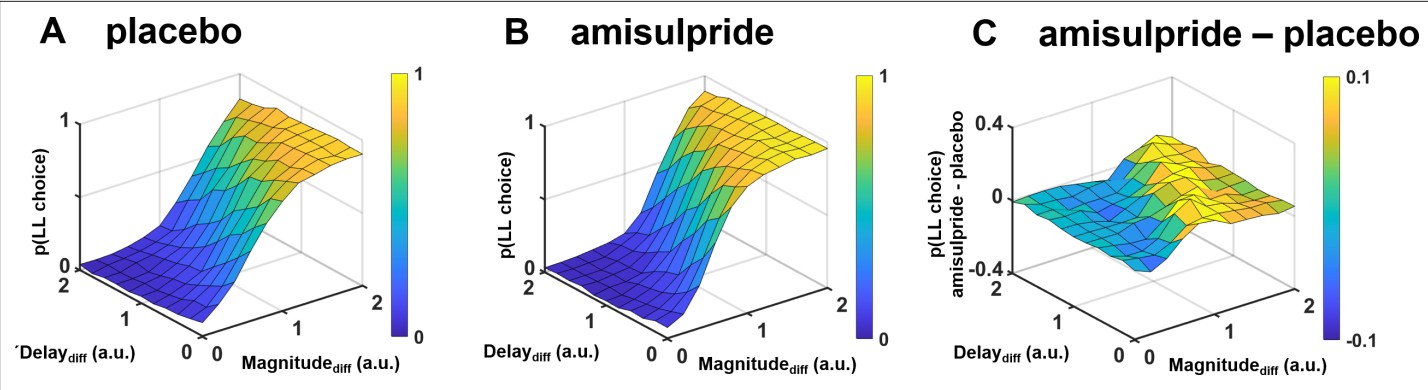

**Figure 5.** Illustration of how dopaminergic effects on intertemporal choices depend on differences in both reward magnitude and delay in the proposed framework, separately for (**A**) placebo, (**B**) amisulpride, and (**C**) the difference between amisulpride and placebo. Plots are based on simulations assuming the group-level parameter estimates we observed under placebo and amisulpride. As dopaminergic effects on decision making affect both reward processing (via the drift rate) and cost processing (via the starting bias), the specific combination of rewards and delays determines whether D2R blockade increases or decreases the probability of larger-later (LL) choices. Low dopamine levels reduce the proximity advantage of smaller-sooner (SS) over LL options particularly if differences in action costs between reward options are large, promoting choices of the LL option. In contrast, if no option possesses a proximity advantage (small differences between delays), dopaminergic effects on evidence accumulation dominate, such that the LL option is perceived as less worth the waiting costs, particularly if its reward magnitude differs only little from that of the alternative SS option.

in favor of low-cost rewards over high-cost rewards. By extension, higher dopamine levels should increase the preference for an SS option if the SS option has a pronounced proximity advantage over the LL option, and bias the acceptance of LL options if both options are associated with similar waiting costs. We note though that the effects of increasing dopamine levels are less predictable than the effects of lowering dopaminergic activity due to possible inverted-U-shaped dopamine-response curves (*Floresco, 2013*); potentially, the dopaminergic effects on drift rate and starting bias might even follow different dose-response functions. Taken together, our process model of the dopaminergic involvement in cost-benefit decisions allows reconciling conflicting theoretical accounts and (apparently) inconsistent empirical findings by showing that dopamine moderates the effects of reward magnitudes and delay costs on different subcomponents of the choice process.

We note that the moderating roles of differences in delays are also reflected in the significant interaction between drug and delay from the model-free analysis, although this analysis could provide no insights into which subcomponents of the choice process are affected by dopamine. As the influence of dopamine on decision making varies as a function of the differences in reward magnitude and waiting costs, the outcomes of standard analyses like mean percentage of LL choices or hyperbolic discount parameters may be specific to the reward magnitudes and delays administered in a given study. For example, if an experimental task includes large differences between rewards and delays, dopamine antagonists may reduce delay discounting, whereas studies with smaller differences between these choice attributes may observe no effect of dopaminergic manipulations (*Figure 5*). Standard analyses that measure patience by one behavioral parameter only (e.g., discount factors) may thus result in misleading findings. In contrast, process models of decision making do not just assess whether a neural manipulation increases or reduces patience; instead, they quantify the influence of a manipulation on the weights assigned to rewards and waiting costs during different phases of the choice process, with these weights being less sensitive to the administered choice options in a given experiment. Process models may thus provide a less option-specific picture of the impact of pharmacological and neural manipulations.

As potential alternative explanation for the enhanced influence of reward magnitude under amisulpride, one might argue that D2R blockade generally increases the signal-to-noise ratio for decision-relevant information. However, this notion is inconsistent with the proposed role of D2R activation for precise action selection (*Keeler et al., 2014*), because this view would have predicted amisulpride to result in noisier (less precise action selection) rather than less noisy evidence accumulation. Moreover, our data provide no evidence for drug effects on the inverse temperature parameter measuring choice consistency, and there were also no significant correlations between amisulpride effects on

reward and delay processing, contrary to what one should expect if these effects were driven by the same mechanism.

While higher doses of amisulpride (as administered in the current study) antagonize postsynaptic D2Rs, lower doses (50–300 mg) were found to primarily block presynaptic dopamine receptors (*Schoemaker et al., 1997*), which may result in amplified phasic dopamine release and thus increased sensitivity to benefits (*Frank and O'Reilly, 2006*). At first glance, the stronger influence of differences in reward magnitude on drift rates under amisulpride compared with placebo might therefore speak in favor of presynaptic (higher dopamine levels) rather than postsynaptic mechanisms of action in the current study. However, amisulpride vs. placebo increased evidence accumulation toward LL rewards (more positive drift rate) only for larger differences between larger (later) and smaller (sooner) rewards, whereas for smaller reward differences amisulpride enhanced evidence accumulation toward SS choices (more negative drift rate; see *Figure 2C*). The latter finding appears inconsistent with presynaptic effects, as higher dopamine levels are thought to increase the preference for costly larger rewards (*Webber et al., 2021*). Instead, the stronger influence of reward differences on drift rates under amisulpride could be explained by a stricter cost control (*Beeler and Mourra, 2018*). In this interpretation, individuals more strongly distinguish between larger rewards that are worth the waiting costs (large difference between LL and SS rewards) and larger rewards that are not worth the same waiting costs (small difference between LL and SS rewards). While this speaks in favor of postsynaptic effects, we acknowledge that the amisulpride effects for larger reward differences are compatible with presynaptic mechanisms.

The result pattern for the starting bias parameter, in turn, suggests the presence of two distinct response biases, reflected by the intercept and the delay-dependent slope of the bias parameter (see *Figure 2K*), which are both under dopaminergic control but in opposite directions. First, participants seem to have a general bias toward the LL option in the current task (intercept), which is reduced under amisulpride compared with placebo, consistent with the assumption that dopamine strengthens the preference for larger rewards (*Beeler and Mourra, 2018*; *Salamone and Correa, 2012*; *Schultz, 2015*). Second, amisulpride reduced the impact of increasing differences in delay on the starting bias, as predicted by the proximity account of tonic dopamine (*Westbrook and Frank, 2018*). Both of these effects are compatible with postsynaptic effects of amisulpride. However, we note that in principle one might make the assumption that proximity effects are stronger for smaller than for larger differences in delay, and under this assumption the results would be consistent with presynaptic effects. On balance, the current results thus appear more likely under the assumption of postsynaptic rather than presynaptic effects but the latter cannot be entirely excluded. Unfortunately, the lack of a significant amisulpride effect on decision times (which should be reduced or increased as consequence of presynaptic or postsynaptic effects, respectively) sheds no additional light on the issue. Lastly, while the actions of amisulpride on D2/D3 receptors are relatively selective, it also affects serotonergic 5-HT7 receptors (*Abbas et al., 2009*). Because serotonin has been related to impulsive behavior (*Mori et al., 2018*), it is worth keeping in mind that amisulpride effects on serotonergic, in addition to dopaminergic, activity might contribute to the observed result pattern.

An important question refers to whether our findings for delay costs can be generalized to other types of costs as well, including risk, social costs (i.e., inequity), effort, and opportunity costs. We recently proposed that dopamine might also moderate proximity effects for reward options differing in risk and social costs, whereas the existing literature provides no evidence for a proximity advantage of effort-free over effortful rewards (*Soutschek et al., 2022*). However, these hypotheses need to be tested more explicitly by future investigations. Dopamine has also been ascribed a role for moderating opportunity costs, with lower tonic dopamine reducing the sensitivity to opportunity costs (*Niv et al., 2007*). While this appears consistent with our finding that amisulpride (under the assumption of postsynaptic effects) reduced the impact of delay on the starting bias, it is important to note that choosing delayed rewards did not involve any opportunity costs in our paradigm, given that participants could pursue other rewards during the waiting time. Thus, it needs to be clarified whether our findings for delayed rewards without experienced waiting time can be generalized to choice situations involving experienced opportunity costs.

To conclude, our findings may shed a new light on the role of dopamine in psychiatric disorders that are characterized by deficits in impulsiveness or cost-benefit weighting in general (*Hasler, 2012*), and where dopaminergic drugs belong to the standard treatments for deficits in value-related and

other behavior. Dopaminergic manipulations yielded mixed results on impulsiveness in psychiatric and neurologic disorders (*Acheson and de Wit, 2008*; *Antonelli et al., 2014*; *Foerde et al., 2016*; *Kayser et al., 2017*), and our process model regarding the role of dopamine for delaying gratification explains some of the inconsistencies between empirical findings (on top of factors like non-linear dose-response relationships). As similarly inconsistent findings were observed also in the domains of risky and social decision making (*Soutschek et al., 2022*; *Webber et al., 2021*), the proposed process model may account for the function of dopamine in these domains of cost-benefit weighting as well. By deepening the understanding of the role of dopamine in decision making, our findings provide insights into how abnormal dopaminergic activation, and its pharmacological treatment, in psychiatric disorders may affect distinct aspects of decision making.

## Materials and methods

### Participants

#### D2 antagonist study

In a double-blind, randomized, within-subject design, 56 volunteers (27 female, $M_{age}$ = 23.2 years, $SD_{age}$ = 3.1 years) received 400 mg amisulpride or placebo in two separate sessions (2 weeks apart) as described previously (*Soutschek et al., 2017*). Participants gave informed written consent before participation. The study was approved by the Cantonal ethics committee Zurich (2012-0568).

#### D1 agonist study

Detailed experimental procedures for the D1 experiment are reported in *Soutschek et al., 2020a*. A total of 120 participants (59 females, mean age = 22.57 years, range 18–28) received either placebo or one of three different doses (6, 15, 30 mg) of the D1 agonist PF-06412562 (between-subject design). The study was approved by the Cantonal ethics committee Zurich (2016-01693) and participants gave informed written consent prior to participation. The D1 agonist study was registered on ClinicalTrials. gov (identifier: NCT03181841).

### Task design

In the D2 antagonist study, participants made intertemporal decisions 90 min after drug or placebo intake. We used a dynamic version of a delay discounting task in which the choice options were individually selected such that the information provided by each decision was optimized (dynamic experiments for estimating preferences; *Toubia et al., 2013*). On each trial, participants decided between an SS (reward magnitude 5–250 Swiss francs, delay 0–30 days) and an LL option (reward magnitude 15–300 Swiss francs, delay 3–90 days). Participants pressed the left or right arrow keys on a standard keyboard to choose the option presented on the left or right side of the screen. On each trial, the reward options were presented until participants made a choice. The next choice options were displayed after an intertrial interval of 1 s. Participants made a total of 20 choices between SS and LL options.

In the D1 agonist experiment, participants performed a task battery including an intertemporal decision task 5 hr after drug administration (the procedures and results for the other tasks are described in *Soutschek et al., 2020b*, and *Soutschek et al., 2020b*). In the intertemporal decision task, the magnitude of the immediate reward option varied between 0 and 16 Swiss francs (in steps of 2 Swiss francs), while for the LL option a fixed amount of 16 Swiss francs was delivered after a variable delay of 0–180 days. A total of 54 trials was administered where each combination of SS and LL reward options was presented once. SS and LL options were randomly presented on either the right or left screen side until a choice was made, and participants indicated their choices by pressing the right arrow key (for the option presented on the right side) or the left arrow key (for the option on the left side).

### Statistical analysis

#### Drift diffusion modeling

We analyzed drug effects on intertemporal decision making with hierarchical Bayesian drift diffusion modeling using the JAGS software package (*Plummer, 2003*). JAGS utilizes Markov Chain Monte Carlo sampling for Bayesian estimation of drift diffusion parameters (drift rate $v$, boundary α, bias $\zeta$, and non-decision time $\tau$) via the Wiener module (*Wabersich and Vandekerckhove, 2014*) on

both the group and the participant level. In our models, the upper boundary (decision threshold) was associated with a choice of the LL option, the lower boundary with a choice of the SS option. A positive drift rate thus indicates evidence accumulation toward the LL option, a negative drift rate toward the SS option. We first describe how the models were set up for the D2 antagonist study. As we were interested in how dopamine modulates different subcomponents of the choice process, in DDM-1 we assumed that the drift rate v is influenced by the comparisons of reward magnitudes and delays between the SS and LL options (*Amasino et al., 2019*; *Dai and Busemeyer, 2014*):

$$\nu' = \beta_1 \left(\text{Magnitude}_{\text{diff}}\right) + \beta_2 \left(\text{Drug} \times \text{Magnitude}_{\text{diff}}\right) + \beta_3 \left(\text{Delay}_{\text{diff}}\right) + \beta_4 \left(\text{Drug} \times \text{Delay}_{\text{diff}}\right) \quad (1)$$

Magnitude$_{\text{diff}}$ indicates the difference between the reward magnitudes of the LL and SS options, Delay$_{\text{diff}}$ indicates the difference between the corresponding delays. Both Magnitude$_{\text{diff}}$ and Delay$_{\text{diff}}$ were z-transformed to render the size of the parameter estimates comparable (*Amasino et al., 2019*). Following previous procedures, we transformed v' with a sigmoidal link function as this procedure explains observed behavior better than linear link functions (*Fontanesi et al., 2019*; *Wagner et al., 2020*). Indeed, also the current data were better explained by a DDM with (DIC =9478) than without (DIC =10,283) a sigmoidal link (where v$_{\text{max}}$ indicates the upper and lower borders of the drift rate):

$$\nu = 2 \times \frac{\beta_5 \left(v_{\text{max}}\right) + \beta_6 \left(\text{Drug} \times v_{\text{max}}\right)}{1 + \exp\left(-v'\right)} - \left(\beta_5 \left(v_{\text{max}}\right) + \beta_6 \left(\text{Drug} \times v_{\text{max}}\right)\right) \quad (2)$$

Next, we assessed whether delay costs affect the starting bias parameter $\zeta$, as assumed by proximity accounts (*Soutschek et al., 2022*; *Westbrook and Frank, 2018*):

$$\zeta = \beta_7 \left(\text{Intercept}\right) + \beta_8 \left(\text{Drug}\right) + \beta_9 \left(\text{Delay}_{\text{diff}}\right) + \beta_{10} \left(\text{Drug} \times \text{Delay}_{\text{diff}}\right) \quad (3)$$

We also investigated whether the drug affected the decision threshold parameter $\alpha$ (*Equation 4*) or the non-decision time $\tau$ (*Equation 5*):

$$\alpha = \beta_{11} \left(\text{Intercept}\right) + \beta_{12} \left(\text{Drug}\right) \quad (4)$$

$$\tau = \beta_{13} \left(\text{Intercept}\right) + \beta_{14} \left(\text{Drug}\right) \quad (5)$$

As the experiment followed a within-subject design, we modeled all parameters both on the group level and on the individual level by assuming that individual parameter estimates are normally distributed around the mean group-level effect with a standard deviation $\lambda$ (which was estimated separately for each group-level effect). We tested for significant effects by checking whether the 95% HDIs of the posterior samples of group-level estimates contained zero. Note that all statistical inferences were based on assessment of group-level estimates, as individual estimates might be less reliable due to the limited number of trials for each participant. We excluded the trials with the 2.5% fastest and 2.5% slowest response times to reduce the impact of outliers on parameter estimation (*Amasino et al., 2019*; *Wagner et al., 2020*). As priors, we assumed standard normal distributions for all group-level effects (with mean = 0 and standard deviation = 1) and gamma distributions for $\lambda$ (*Wagner et al., 2020*). For model estimation, we computed two chains with 500,000 samples (burning = 450,000, thinning = 5). R was used to assess model convergence in addition to visual inspection of chains. For all effects, R was below 1.01, indicating model convergence.

We compared DDM-1 also with alternative process models. DDM-2 was identical to DDM-1 but did not estimate starting bias as free parameter, assuming $\zeta$ =0.5 instead, whereas DDM-3 left out the influences of Magnitude$_{\text{diff}}$ and Delay$_{\text{diff}}$ on the drift rate. DDM-4 assessed whether the starting bias is modulated by the sum of the delays (as measure of overall proximity, Delay$_{\text{sum}}$) rather than Delay$_{\text{diff}}$. In DDM-5 we assumed that the drift rate depends on the comparison of the hyperbolically discounted subjective values of the two choice options rather than on the comparison of choice attributes (*Konovalov and Krajbich, 2019*). In particular, the drift rate $v'$ (prior to being passed through the sigmoidal link function) was calculated with:

$$v' = \frac{LL \; reward \; magnitude}{1 + \left(\beta_1 + \beta_2 \left(Drug\right)\right) \times LL \; delay} - \frac{SS \; reward \; magnitude}{1 + \left(\beta_1 + \beta_2 \left(Drug\right)\right) \times SS \; delay} \quad (6)$$

Here, β$_1$ corresponds to the hyperbolic discount factor, which determines the hyperbolically discounted subjective values of the available choice options.

Finally, we considered a model without influence of Delay$_{diff}$ on the starting bias but with separate non-decision times for rewards and delays. In more detail, DDM-6 included an additional parameter $\tau_{diff}$ which indicated whether the accumulation process started earlier for delays than for rewards ($\tau_{diff} > 0$) or vice versa ($\tau_{diff} < 0$). For example, if $\tau_{diff} > 0$, evidence accumulation for delays starts directly after the non-decision time $\tau$, whereas the accumulation process for reward magnitudes starts at $\tau + \tau_{diff}$ (and then influences the drift rate together with Delay$_{diff}$ until the decision boundary is reached). A recent study showed that such time-varying drift rates can be calculated as follows (**Lombardi and Hare, 2021**):

$$
v' = \begin{cases}
\beta_1(M_{diff}) + \beta_2(Drug \times M_{diff}) & if \tau_{diff} < 0 \ \& \ \tau_{diff} + \tau < RT \\
\beta_3(D_{diff}) + \beta_4(Drug \times D_{diff}) & if \tau_{diff} > 0 \ \& \ \tau_{diff} + \tau < RT \\
\frac{\tau + \tau_{diff}}{RT - \tau + \tau_{diff}} \times (\beta_1(M_{diff}) + \beta_2(Drug \times M_{diff})) + \frac{RT - \tau}{RT - \tau + \tau_{diff}} \times (\beta_1(M_{diff}) + \beta_2(Drug \times M_{diff}) + \beta_3(D_{diff}) + \beta_2(Drug \times D_{diff})) & if \tau_{diff} < 0 \ \& \ \tau < RT \\
\frac{\tau_{diff}}{RT - \tau} \times (\beta_3(D_{diff}) + \beta_4(Drug \times D_{diff})) + \frac{RT - \tau - \tau_{diff}}{RT - \tau} \times (\beta_1(M_{diff}) + \beta_2(Drug \times M_{diff}) + \beta_3(D_{diff}) + \beta_2(Drug \times D_{diff})) & if \tau_{diff} > 0 \ \& \ \tau_{diff} + \tau < RT
\end{cases}
\tag{7}
$$

For the ease of reading, Magnitude$_{diff}$ and Delay$_{diff}$ are abbreviated as M$_{diff}$ and D$_{diff}$, respectively.

For the D1 agonist study, we computed the same DDMs as for the D2 antagonist study. However, because the D1 agonist experiment followed a between-subject design, we estimated separate group-level parameters for the four between-subject drug groups (placebo, 6, 15, 30 mg). We tested for significant group differences by computing the 95% HDI for the differences between the posterior samples of group-level estimates. For model estimation, we computed two chains with 100,000 samples (burning = 50,000, thinning = 5), which ensured that R values for all group-level effects were below 1.01.

We compared model fits between the different DDMs with the deviance information criterion (DIC) as implemented in the Rjags package. We note that JAGS does not allow computing more recently developed model selection criteria such as the Pareto smoothed importance sampling leave-one-out (PSIS-LOO) approach. However, a recent comparison of model selection approaches found that PSIS-LOO had a slightly higher false detection rate than DIC, but in general both PSIS-LOO and DIC led to converging conclusions (**Lu et al., 2017**). There is therefore good reason to assume that our findings were not biased by the employed model selection approach.

## Posterior predictive checks and parameter recovery analyses

We performed posterior predictive checks to assess whether the DDMs explained key aspects of the empirical data. For this purpose, we simulated 1000 RT distributions based on the individual parameter estimates from all DDMs. We then binned trials into quintiles based on differences in reward magnitude and plotted the observed empirical data and the simulated data (averaged across the 1000 simulations) as a function of these bins, separately for each individual participant. We performed the same analysis by binning trials based on differences in delay instead of reward magnitude.

We conducted a parameter recovery analysis by re-computing DDM-1 on 10 randomly selected data sets which were simulated based on the original DDM-1 parameters. We checked parameter recovery by assessing whether group-level parameters from the simulated data lie within the 95% HDI of the original parameter estimates.

## Model-free analyses

We analyzed choice data also in a model-free manner and with a hyperbolic discounting model. In the model-free analysis of the D2 antagonist study, we regressed choices of LL vs. SS options on fixed-effect predictors for Drug, Magnitude$_{diff}$, Delay$_{diff}$, and the interaction terms using Bayesian mixed models as implemented in the brms package in R (**Bürkner, 2017**). For the D1 agonist study, the same MGLM was used with the only difference that Drug (0, 6, 15, 30 mg) represented a between- rather than a within-subject factor. All predictors were also modeled as random slopes in addition to participant-specific random intercepts. Finally, the hyperbolic discounting model was fit using the hBayesDM toolbox (**Ahn et al., 2017**), using a standard hyperbolic discounting function:

$$
SV_{discounted} = \frac{reward\ magnitude}{1 + k \times delay}
\tag{8}
$$

To translate subjective value into choices, we fitted a standard softmax function to each participant's choices:

$$P\left(choice\ of\ LL\ option\right) = \frac{1}{1+e^{-\beta_{temp} \times (SV_{LL} - SV_{SS})}} \tag{9}$$

We estimated parameters capturing the strength of hyperbolic discounting (k) and choice consistency ($\beta_{temp}$) separately for each participant and experimental session by computing two chains of 4000 iterations (burning = 2000). We then performed a Bayesian t-test on the log-transformed individual parameter estimates under placebo vs. amisulpride using the BEST package (*Kruschke, 2013*).

## Acknowledgements

PNT received funding from the Swiss National Science Foundation (Grants 100019_176016, 100014_165884, and CRSII5_177277) and from the Velux Foundation (Grant 981). AS received an Emmy Noether fellowship (SO 1636/2-1) from the German Research Foundation.

## Additional information

### Funding

| Funder | Grant reference number | Author |
|---|---|---|
| Deutsche Forschungsgemeinschaft | SO 1636/2-1 | Alexander Soutschek |
| Swiss National Science Foundation | 100019_176016 | Philippe N Tobler |
| Velux Stiftung | 981 | Philippe N Tobler |
| Swiss National Science Foundation | 100014_165884 | Philippe N Tobler |
| Swiss National Science Foundation | CRSII5_177277 | Philippe N Tobler |

The funders had no role in study design, data collection and interpretation, or the decision to submit the work for publication.

### Author contributions

Alexander Soutschek, Conceptualization, Formal analysis, Investigation, Visualization, Methodology, Writing - original draft; Philippe N Tobler, Conceptualization, Supervision, Funding acquisition, Methodology, Writing – review and editing

### Author ORCIDs

Alexander Soutschek http://orcid.org/0000-0001-8438-7721
Philippe N Tobler http://orcid.org/0000-0002-4915-9448

### Ethics

Clinical trial registration The D1 agonist study was registered on ClinicalTrials.gov (identifier: NCT03181841).
Human subjects: Participants gave informed written consent before participation. The Cantonal ethics committee Zurich approved both the D2 antagonist study (2012-0568) and the D1 agonist study (2016-01693).

### Decision letter and Author response

Decision letter https://doi.org/10.7554/eLife.83734.sa1
Author response https://doi.org/10.7554/eLife.83734.sa2

## Additional files

### Supplementary files
• MDAR checklist

### Data availability

The data supporting the findings of this study and the data analysis code are available on Open Science Framework (https://osf.io/dp2me/).

The following dataset was generated:

| Author(s) | Year | Dataset title | Dataset URL | Database and Identifier |
|---|---|---|---|---|
| Soutschek A | 2023 | Intertemporal Choice_amisulpride | https://osf.io/dp2me/ | Open Science Framework, dp2me |

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
