## [Editor Report]

This important study reanalyzes a prior dataset testing effects of D2 antagonism on choices in a delay discounting task. While the prior report using standard analysis, showed no effects, the current study used a DDM to examine more carefully possible contrasting effects on different subcomponents of the decision process. This approach revealed convincing evidence of contrasting effects with D2 blockade increasing the effect of reward size differences to favor selection of the larger, later reward, while also shifting the bias toward selection of the small immediate reward. The authors speculate that these opposing effects explain the variability in effects across studies, since they mean that effects would depend on which of these factors is more important in a particular design.

---

## [Decision Letter]

**Decision letter after peer review:**

Thank you for submitting your article "A Process Model Account of the Role of Dopamine in Intertemporal Choice" for consideration by *eLife*. Your article has been reviewed by 3 peer reviewers, including Geoffrey Schoenbaum as Reviewing Editor and Reviewer #1, and the evaluation has been overseen by Christian Büchel as the Senior Editor.

Essential revisions:

Overall the reviewers thought the work had great merit and was well done. The conclusions are novel and provide potential resolution of conflicting data in the literature. However, there were a number of concerns raised.

Many of these suggestions are contained in each reviewers comments, however on discussion we reached a general consensus that there were three main areas that require some additional analysis and/or changes to the framing or interpretation.

1) The first is the generalization from a single task or paradigm – delay discounting – to broader conclusions about cost evaluation. Temporal discounting is obviously just one factor in value or costs. Comments or discussion of this and the likelihood or lack thereof that conclusions will generalize outside the realm of time would be appreciated.

2) The second is similar and it concerns limitations on the use of a receptor specific pharmacological agent. As indicated in the review, it is somewhat unclear if this impacts phasic or tonic functions of dopamine or both, and of course it leaves open the role of other receptor systems as well as the locus of action – pre vs post-synaptic. Acknowledging these limitations and commenting on whether there is any insight into them would also be an improvement.

3) Finally it was suggested the authors might look for other similar studies and/or publicly available data that could be similarly analyzed. If so, applying the current model to a data from a different type of cost/benefit choice task or to data from a discounting task where a different pharmacological agent was used would greatly expand the potential impact of the conclusions and perhaps allow points 1 and 2 to be directly addressed.

*Reviewer #1 (Recommendations for the authors):*

I have a few comments however.

One is that I wonder if the authors could frame the results more in terms of current proposed functions of dopamine. I gather this is nothing related to error signaling, learning functions, or other effects of phasic activity. That is, the current ideas apply – appropriately since the evidence is pharmacological – to the role of tonic dopamine levels? Can this be more clearly stated? What is the evidence that these factors are not affected?

Second the antagonist is D2 specific. This is not much mentioned. How important is this? What do studies of D1 or general antagonism find?

Third would the effects here be explained by the more general hypothesis that tonic dopamine relates to the value of time? I refer to a proposal made by Yael Niv among others. It seems to me that blocking dopamine would in effect lower the value of time, which would be expected to impact the measures described here, for example by making a subject more willing to wait across a delay? Could this idea be related to the findings?

*Reviewer #2 (Recommendations for the authors):*

Overall, I appreciated the detailed modeling work, and the thoughtful alternative constructions of the models. Given this, I wanted to see more of the data and results myself – more informative plots that show both main features of the behavior, and more detailed visual presentation of the actual posterior estimates themselves. I think it would also be useful to plot model comparisons – both globally and for individuals – these give a sense of how close different models are. It was hard to keep track of all the different parameters too so a table of the model parameters for each flavor DDM would be extremely useful for understanding and transparency.

I did not find the posterior predictive check particularly convincing – the authors say that the DDMs are a reasonable account, but from purely visual assessment I can see plenty of individual subjects in which the models (all of the variants) are not an unambiguously good approximation. Some kind of summary statistic of the posterior predictive check might be more interpretable?

*Reviewer #3 (Recommendations for the authors):*

It is possible that the Authors could gain some inferential leverage by examining the effect of drug on reaction times alone. If amisulpride increases post-synaptic dopamine signaling, we might see faster reaction times overall. We also might see reaction times speed when Participants are offered larger overall valued offer pairs and this effect might be larger on amisulpride versus placebo. In any case, it would be helpful for the Authors to report RT effects of the drug. What were the main effects of the drug on reaction times? What were the marginal effects of the drug on reaction times, controlling for differences in reward amount, delay, and choice, etc.?

It is also possible that there are trial-wise dynamics which may be informative. I am curious whether the Authors examined trial-number effects on the propensity to select the LL option. If the propensity either grew or shrank over trials, then it may be possible to test whether possible post-synaptic dopamine signaling amplified this bias towards the LL option when it was smaller versus larger.

[Editors’ note: further revisions were suggested prior to acceptance, as described below.]

Thank you for resubmitting your work entitled "A Process Model Account of the Role of Dopamine in Intertemporal Choice" for further consideration by *eLife*. Your revised article has been evaluated by Christian Büchel (Senior Editor) and Geoffrey Schoenbaum (Reviewing Editor).

The manuscript has been improved but there are some remaining issues that need to be addressed. Specifically, Reviewer 3 disagrees with some of the interpretations being advanced. We wish to give you an opportunity to make any further modifications to your manuscript/rebuttal in light of these comments in a final revision.

*Reviewer #1 (Recommendations for the authors):*

Thanks for addressing all my concerns - no further suggestions!

*Reviewer #2 (Recommendations for the authors):*

The amendments to the paper address all my previous comments and have improved the manuscript overall.

*Reviewer #3 (Recommendations for the authors):*

The Authors have been responsive to my prior comments. The test of the model in a sample with D1 agonists is welcome and informative – if disappointingly null. In general, I think the Author's edits are suitable.

I disagree with the Authors' response about how the drug effects on the starting point bias and the effects of costs should be interpreted. I still think the data tell a more consistent story, and coincidentally one that is also more consistent with the Westbrook and Frank (2018) hypothesis, if interpreted under the assumption of pre- rather than post-synaptic effects.

Before giving my reasoning, I want to be clear that I think the Authors' interpretation is plausible under narrower assumptions. Also, it is good that they have not ruled out the possibility of pre-synaptic drug effects. If anything, I think they could make a bit more clear in the discussion that an alternative, presynaptic account is plausible (and why this would also support DA-proximity interactions). Ultimately, though, I think this is up to the Authors' discretion. As such, I wanted to articulate my reasoning here in case I might persuade them to reconsider the inferences they make.

There are two points on which I disagree with the Authors' response:

1) the effect of amisulpride on the magnitude effect and 2) drug effects on the DDM starting point bias.

Regarding (1): contrary to the Authors' take, Figure 2B indicates that amisulpride vs. placebo increased the effect of reward magnitude on choice, which supports a pre-synaptic effect. 2C further supports this pre-synaptic account because the drift rate also appears to steepen (as a function of differences in reward magnitude) on amisulpride versus placebo. That is – both figures support a larger reward magnitude/benefit effect on choice, consistent with pre-synaptic amisulpride action, yielding stronger post-synaptic dopamine signaling. In contrast, there is no reliable interaction between drug and delay (Figure 2E) which contradicts the Authors' assertions that D2R activation is implementing a "cost control".

Regarding (2): I'm also inclined to disagree with the Authors' interpretation of the drug effect on the DDM starting bias. While an overall starting bias, and its variation by delay are reduced on amisulpride, the key theoretical question is whether amisulpride alters the proximity bias (stemming from the differences in delay), not the DDM starting bias (though, theoretically, the proximity bias may be reflected in a DDM starting bias).

As noted in my prior review, the drug effect on the DDM starting point may reflect either a reduction of a tendency to select the LL option, or an increase in a proximity bias (arising from differences in delay), or both. I think we are in agreement that all three are possible and that it is not possible to discern between these accounts. Evidence that a proximity bias exists includes that the DDM starting point shifts toward the SS threshold as the differences in delay increase (Figure 2K, on placebo).

The original theory proposes that greater DA signaling increases the SS proximity bias (which predicts a shift in the DDM starting point towards the SS boundary). Contrary to the Authors' claims, there is a main effect of the drug, shifting the DDM starting point towards the SS boundary (Figure 2H). Again, this could be explained by a weakening of the tendency to select the LL option, a strengthening of the proximity bias towards the SS option, or both. Any of these effects could explain Figure 2H. Importantly, however, Figure 2H is inconsistent with the interpretation that "amisulpride weakens the proximity advantage of SS over LL rewards".

Regarding Figure 2K, it is unclear why presynaptic amisulpride binding would strengthen the proximity bias mostly for smaller differences in delay. Nevertheless, the fact that the drug caused a shift in the starting point towards the SS option more for small versus large differences in delay does not contradict the hypothesis that amisulpride pre-synaptically causes a shift in the starting point towards SS rewards. It merely implies that the effects were largest when delay differences were smallest. Perhaps this is true because there is a ceiling effect on how much the proximity bias can influence choice in this paradigm, and it's already maxed out (on placebo) for large delay differences such that the drug can only amplify the proximity bias for small delay differences.

---

## [Author Response]

Essential revisions:Overall the reviewers thought the work had great merit and was well done. The conclusions are novel and provide potential resolution of conflicting data in the literature. However there were a number of concerns raised.Many of these suggestions are contained in each reviewers comments, however on discussion we reached a general consensus that there were three main areas that require some additional analysis and/or changes to the framing or interpretation.

We thank the reviewers for the positive evaluation of our study. Revising the manuscript according to the reviewers’ comments further improved the quality of the manuscript.

1) The first is the generalization from a single task or paradigm – delay discounting – to broader conclusions about cost evaluation. Temporal discounting is obviously just one factor in value or costs. Comments or discussion of this and the likelihood or lack thereof that conclusions will generalize outside the realm of time would be appreciated.

We agree that temporal delay is just one cost factor influencing value. In the revised manuscript, we clarify that one must be cautious with generalizing our findings for delay discounting to other cost types (for details, see responses to reviewers 1 and 2).

2) The second is similar and it concerns limitations on the use of a receptor specific pharmacological agent. As indicated in the review, it is somewhat unclear if this impacts phasic or tonic functions of dopamine or both, and of course it leaves open the role of other receptor systems as well as the locus of action – pre vs post-synaptic. Acknowledging these limitations and commenting on whether there is any insight into them would also be an improvement.

We added a discussion on whether the observed amisulpride effects can best be explained by presynaptic or postsynaptic effects. We clarify that amisulpride is thought to modulate D2R-mediated tonic dopamine levels, but it may also affect 5-HT7 receptors (see response to reviewer 2).

3) Finally it was suggested the authors might look for other similar studies and/or publicly available data that could be similarly analyzed. If so, applying the current model to a data from a different type of cost/benefit choice task or to data from a discounting task where a different pharmacological agent was used would greatly expand the potential impact of the conclusions and perhaps allow points 1 and 2 to be directly addressed.

We thank the reviewers or this interesting suggestion! In the revised manuscript, we now reported data from a study on the impact of a D1 agonist on intertemporal choice. Conducting the same analyses on this data set as for the D2 antagonist study yielded no significant effects of D1R stimulation on drift diffusion parameters. Based on these findings, it seems more likely that the impact of amisulpride on the choice process is moderated by effects on D2R rather than D1R activation.

Reviewer #1 (Recommendations for the authors):I have a few comments however.One is that I wonder if the authors could frame the results more in terms of current proposed functions of dopamine. I gather this is nothing related to error signaling, learning functions, or other effects of phasic activity. That is, the current ideas apply – appropriately since the evidence is pharmacological – to the role of tonic dopamine levels? Can this be more clearly stated? What is the evidence that these factors are not affected?

In the revised manuscript, we now clarify that amisulpride is thought to affect mainly tonic dopaminergic activity, given the half-life of amisulpride (about 12 hours) and given that D2 receptors are more susceptible to slow, tonic changes in dopamine concentrations rather than to fast, phasic dopaminergic bursts. Tonic dopaminergic activity is thought to encode background reward rates, in contrast to the learning processes mediated by phasic dopamine (as described by the reviewer). We now clarify this on p.5:

“D1Rs are prevalent in the direct “Go” pathway and facilitate action selection via mediating the impact of phasic bursts elicited by high above-average rewards (Evers, Stiers, and Ramaekers, 2017; Kirschner, Rabinowitz, Singer, and Dagher, 2020). D2Rs, in contrast, dominate the indirect “Nogo” pathway (which suppresses action) and are more sensitive to small concentration differences in tonic dopamine levels (Missale, Nash, Robinson, Jaber, and Caron, 1998), which is thought to encode the background, average reward rate (Kirschner et al., 2020; Volkow and Baler, 2015; Westbrook and Frank, 2018; Westbrook et al., 2020).”

Second the antagonist is D2 specific. This is not much mentioned. How important is this? What do studies of D1 or general antagonism find?

We agree that the receptor specificity of our findings is an important issue. While we are not aware of a publically available dataset using a D1R antagonist, we now report findings from a D1 agonist study (results from this study were already published in Soutschek et al., 2020, *Biological Psychiatry*). A re-analysis of this data set (i.e,, the same models as for the D2 antagonist study) provided no evidence for an influence of D1R stimulation on drift diffusion parameters. It seems thus more likely that the amisulpride effects on the decision process are moderated via D2R rather than D1R activity.

The findings for the D1 agonist study are reported in the revised manuscript on p.16:

“To assess the receptor specificity of our findings, we conducted the same analyses on the data from a study (published previously in Soutschek et al. (2020)) testing the impact of three doses of a D1 agonist (6 mg, 15 mg, 30 mg) relative to placebo on intertemporal choices (between-subject design). In the intertemporal choice task used in this experiment, the SS reward was always immediately available (delay = 0), contrary to the task in the D2 experiment where the delay of the SS reward varied from 0-30 days. Again, the data in the D1 experiment were best explained by DDM-1 (DIC_DDM-1_ = 19,657) compared with all other DDMs (DIC_DDM-2_ = 20,934; DIC_DDM-3_ = 21,710; DIC_DDM-5_ = 21,982; DIC_DDM-6_ = 19,660; note that DDM-4 was identical with DDM-1 for the D1 agonist study because the delay of the SS reward was 0). Neither the best-fitting nor any other model yielded significant drug effects on any drift diffusion parameter (see Table 4 for the best-fitting model). Also model-free analyses conducted in the same way as for the D2 antagonist study revealed no significant drug effects (all HDI_95%_ included zero). There was thus no evidence for any influence of D1R stimulation on intertemporal decisions.”

We added a discussion of the receptor specificity of our findings on p.17:

“This finding represents first evidence for the hypothesis that tonic dopamine moderates the impact of proximity (e.g., more concrete versus more abstract rewards) on cost-benefit decision making (Soutschek, Jetter, and Tobler, 2022; Westbrook and Frank, 2018). Pharmacological manipulation of D1R activation, in contrast, showed no significant effects on the decision process. This provides evidence for the receptor specificity of dopamine’s role in intertemporal decision making (though as caveat it is worth keeping the differences between the tasks administered in the D1 and the D2 studies in mind).”

Third would the effects here be explained by the more general hypothesis that tonic dopamine relates to the value of time? I refer to a proposal made by Yael Niv among others. It seems to me that blocking dopamine would in effect lower the value of time, which would be expected to impact the measures described here, for example by making a subject more willing to wait across a delay? Could this idea be related to the findings?

We agree that it is important to relate our current findings to the hypothesis that dopamine encodes the opportunity costs of actions (Niv et al., 2007). Opportunity costs occur in tasks where participants actually experience the waiting time for a reward and cannot engage in other goal-directed behaviors during this waiting time. The intertemporal decision task we used in our study is virtually free from any opportunity costs (other than those associated with taking part in the experiment), as participants could engage in other behaviors while waiting for the delivery of delayed rewards. Thus, while our finding that D2R blockade reduces the impact of delay costs on the starting bias appears to be consistent with Niv’s hypothesis that blocking dopamine lowers the sensitivity to opportunity (waiting) costs, future studies may have to test whether dopamine modulates also the impact of experienced waiting costs on the starting bias. We discuss this issue on p.22:

“Dopamine has also been ascribed a role of moderating opportunity costs, with lower tonic dopamine reducing the sensitivity to opportunity costs (Niv, Daw, Joel, and Dayan, 2007). While this appears consistent with our finding that amisulpride (under the assumption of postsynaptic effects) reduced the impact of delay on the starting bias, it is important to note that choosing delayed rewards did not involve any opportunity costs in our paradigm, given that participants could pursue other rewards during the waiting time. Thus, it needs to be clarified whether our findings for delayed rewards without experienced waiting time can be generalized to choice situations involving experienced opportunity costs.”

Reviewer #2 (Recommendations for the authors):Overall, I appreciated the detailed modeling work, and the thoughtful alternative constructions of the models. Given this, I wanted to see more of the data and results myself – more informative plots that show both main features of the behavior, and more detailed visual presentation of the actual posterior estimates themselves. I think it would also be useful to plot model comparisons – both globally and for individuals – these give a sense of how close different models are. It was hard to keep track of all the different parameters too so a table of the model parameters for each flavor DDM would be extremely useful for understanding and transparency.

We added further plots visualizing the main features of behavior as well as modelling results. We added plots showing model-free results (Figure 1C/D) as well as the posterior parameter distributions for the influences of reward and delay on the drift rate or the starting bias (Figure 2). We also added a figure illustrating the model comparisons (Figure 3A). Lastly, we added a table providing an overview over the parameters included in the different DDMs (Table 3).

I did not find the posterior predictive check particularly convincing – the authors say that the DDMs are a reasonable account, but from purely visual assessment I can see plenty of individual subjects in which the models (all of the variants) are not an unambiguously good approximation. Some kind of summary statistic of the posterior predictive check might be more interpretable?

We followed the reviewer’s recommendation and added summary statistics for the posterior predictive checks by plotting the empirically observed and simulated reaction times collapsed across all participants (Figure 3B/C). The summary statistics reveal that at least DDMs 1-3 provide good explanations for the empirically observed data, whereas DDMs 4-6 more strongly deviated from the observed behavior.

Reviewer #3 (Recommendations for the authors):It is possible that the Authors could gain some inferential leverage by examining the effect of drug on reaction times alone. If amisulpride increases post-synaptic dopamine signaling, we might see faster reaction times overall. We also might see reaction times speed when Participants are offered larger overall valued offer pairs and this effect might be larger on amisulpride versus placebo. In any case, it would be helpful for the Authors to report RT effects of the drug. What were the main effects of the drug on reaction times? What were the marginal effects of the drug on reaction times, controlling for differences in reward amount, delay, and choice, etc.?

We thank the reviewer for this interesting suggestion. We conducted a Bayesian mixed model regressing log-transformed reaction times on predictors for Drug, Choice, Magnitude_diff_, Delay_diff_, and the sum of the reward magnitudes (Magnitude_sum_). As predicted by the reviewer, reaction times were faster if offers included larger overall reward magnitudes, HDI_mean_ = -0.12, HDI_95%_ = [-0.18; -0.06], but neither this nor any other effect was significantly modulated by amisulpride (all HDI_95%_ include zero). The decision time data therefore do not allow resolving the question of whether dopamine showed presynaptic (which would predict faster RTs) or postsynaptic (resulting in slower RTs) mechanisms of actions in the current study. We now report the reaction time analysis in the revised manuscript on p.6-7:

“We also tested for drug effects on decision times by re-computing the MGLM reported above on log-transformed decision times, adding predictors for choice (SS versus LL option) and Magnitude_sum_ (combined magnitudes of SS and LL rewards). Participants made faster decisions the higher the sum of the two rewards, HDI_mean_ = -0.12, HDI_95%_ = [-0.18; -0.06], however we observed no significant drug effects on decision times.”

It is also possible that there are trial-wise dynamics which may be informative. I am curious whether the Authors examined trial-number effects on the propensity to select the LL option. If the propensity either grew or shrank over trials, then it may be possible to test whether possible post-synaptic dopamine signaling amplified this bias towards the LL option when it was smaller versus larger.

Following the reviewer’s recommendation, we added trial number (z-transformed) as additional factor to the Bayesian mixed model on choices. Participants made indeed more LL choices with increasing trial number, 95% HDI = [0.19; 0.99], but there was no evidence that amisulpride moderated this influence of trial number on choices, 95% HDI = [-0.61; 0.48]. We report this analysis in the revised manuscript on p.6:

“When we explored whether dopaminergic effects changed over the course of the experiment, we observed a significant main effect of trial number (more LL choices over time), HDI_mean_ = 0.58, HDI_95%_ = [0.19; 0.99]. However, this effect was unaffected by the pharmacological manipulation, HDI_mean_ = -0.06, HDI_95%_ = [-0.61; 0.48].”

References:

Beeler, J. A., and Mourra, D. (2018). To do or not to do: dopamine, affordability and the economics of opportunity. *Frontiers in integrative neuroscience, 12*, 6.

Evers, E., Stiers, P., and Ramaekers, J. (2017). High reward expectancy during methylphenidate depresses the dopaminergic response to gain and loss. *Soc Cogn Affect Neurosci, 12*(2), 311-318.

Frank, M. J., and O'Reilly, R. C. (2006). A mechanistic account of striatal dopamine function in human cognition: psychopharmacological studies with cabergoline and haloperidol. *Behavioral neuroscience, 120*(3), 497.

Kirschner, M., Rabinowitz, A., Singer, N., and Dagher, A. (2020). From apathy to addiction: Insights from neurology and psychiatry. *Progress in Neuro-Psychopharmacology and Biological Psychiatry, 101*, 109926.

Missale, C., Nash, S. R., Robinson, S. W., Jaber, M., and Caron, M. G. (1998). Dopamine receptors: from structure to function. *Physiological reviews, 78*(1), 189-225.

Niv, Y., Daw, N. D., Joel, D., and Dayan, P. (2007). Tonic dopamine: opportunity costs and the control of response vigor. *Psychopharmacology (Berl), 191*(3), 507-520. doi:10.1007/s00213-006-0502-4

Salamone, J. D., and Correa, M. (2012). The mysterious motivational functions of mesolimbic dopamine. *Neuron, 76*(3), 470-485. doi:10.1016/j.neuron.2012.10.021

Schoemaker, H., Claustre, Y., Fage, D., Rouquier, L., Chergui, K., Curet, O.,... Benavides, J. (1997). Neurochemical characteristics of amisulpride, an atypical dopamine D2/D3 receptor antagonist with both presynaptic and limbic selectivity. *Journal of Pharmacology and Experimental Therapeutics, 280*(1), 83-97.

Schultz, W. (2015). Neuronal Reward and Decision Signals: From Theories to Data. *Physiol Rev, 95*(3), 853-951. doi:10.1152/physrev.00023.2014

Soutschek, A., Gvozdanovic, G., Kozak, R., Duvvuri, S., de Martinis, N., Harel, B.,... Tobler, P. N. (2020). Dopaminergic D1 Receptor Stimulation Affects Effort and Risk Preferences. *Biol Psychiatry, 87*(7), 678-685. doi:10.1016/j.biopsych.2019.09.002

Soutschek, A., Jetter, A., and Tobler, P. N. (2022). Towards a Unifying Account of Dopamine’s Role in Cost-Benefit Decision Making. *Biological Psychiatry Global Open Science*.

Volkow, N. D., and Baler, R. D. (2015). NOW vs LATER brain circuits: implications for obesity and addiction. *Trends in neurosciences, 38*(6), 345-352.

Westbrook, A., and Frank, M. (2018). Dopamine and Proximity in Motivation and Cognitive Control. *Curr Opin Behav Sci, 22*, 28-34. doi:10.1016/j.cobeha.2017.12.011

Westbrook, A., van den Bosch, R., Maatta, J. I., Hofmans, L., Papadopetraki, D., Cools, R., and Frank, M. J. (2020). Dopamine promotes cognitive effort by biasing the benefits versus costs of cognitive work. *Science, 367*(6484), 1362-1366. doi:10.1126/science.aaz5891

[Editors’ note: further revisions were suggested prior to acceptance, as described below.]

Reviewer #3 (Recommendations for the authors):The Authors have been responsive to my prior comments. The test of the model in a sample with D1 agonists is welcome and informative – if disappointingly null. In general, I think the Author's edits are suitable.I disagree with the Authors' response about how the drug effects on the starting point bias and the effects of costs should be interpreted. I still think the data tell a more consistent story, and coincidentally one that is also more consistent with the Westbrook and Frank (2018) hypothesis, if interpreted under the assumption of pre- rather than post-synaptic effects.Before giving my reasoning, I want to be clear that I think the Authors' interpretation is plausible under narrower assumptions. Also, it is good that they have not ruled out the possibility of pre-synaptic drug effects. If anything, I think they could make a bit more clear in the discussion that an alternative, presynaptic account is plausible (and why this would also support DA-proximity interactions). Ultimately, though, I think this is up to the Authors' discretion. As such, I wanted to articulate my reasoning here in case I might persuade them to reconsider the inferences they make.There are two points on which I disagree with the Authors' response:1) the effect of amisulpride on the magnitude effect and 2) drug effects on the DDM starting point bias.Regarding (1): contrary to the Authors' take, Figure 2B indicates that amisulpride vs. placebo increased the effect of reward magnitude on choice, which supports a pre-synaptic effect. 2C further supports this pre-synaptic account because the drift rate also appears to steepen (as a function of differences in reward magnitude) on amisulpride versus placebo. That is – both figures support a larger reward magnitude/benefit effect on choice, consistent with pre-synaptic amisulpride action, yielding stronger post-synaptic dopamine signaling. In contrast, there is no reliable interaction between drug and delay (Figure 2E) which contradicts the Authors' assertions that D2R activation is implementing a "cost control".Regarding (2): I'm also inclined to disagree with the Authors' interpretation of the drug effect on the DDM starting bias. While an overall starting bias, and its variation by delay are reduced on amisulpride, the key theoretical question is whether amisulpride alters the proximity bias (stemming from the differences in delay), not the DDM starting bias (though, theoretically, the proximity bias may be reflected in a DDM starting bias).As noted in my prior review, the drug effect on the DDM starting point may reflect either a reduction of a tendency to select the LL option, or an increase in a proximity bias (arising from differences in delay), or both. I think we are in agreement that all three are possible and that it is not possible to discern between these accounts. Evidence that a proximity bias exists includes that the DDM starting point shifts toward the SS threshold as the differences in delay increase (Figure 2K, on placebo).The original theory proposes that greater DA signaling increases the SS proximity bias (which predicts a shift in the DDM starting point towards the SS boundary). Contrary to the Authors' claims, there is a main effect of the drug, shifting the DDM starting point towards the SS boundary (Figure 2H). Again, this could be explained by a weakening of the tendency to select the LL option, a strengthening of the proximity bias towards the SS option, or both. Any of these effects could explain Figure 2H. Importantly, however, Figure 2H is inconsistent with the interpretation that "amisulpride weakens the proximity advantage of SS over LL rewards".Regarding Figure 2K, it is unclear why presynaptic amisulpride binding would strengthen the proximity bias mostly for smaller differences in delay. Nevertheless, the fact that the drug caused a shift in the starting point towards the SS option more for small versus large differences in delay does not contradict the hypothesis that amisulpride pre-synaptically causes a shift in the starting point towards SS rewards. It merely implies that the effects were largest when delay differences were smallest. Perhaps this is true because there is a ceiling effect on how much the proximity bias can influence choice in this paradigm, and it's already maxed out (on placebo) for large delay differences such that the drug can only amplify the proximity bias for small delay differences.

We agree with the reviewer that, depending on the underlying assumptions, the current result pattern can be explained by both postsynaptic and presynaptic mechanisms. Accordingly, we added a more balanced discussion of the arguments in favor of pre- versus postsynaptic effects.

Regarding the drug effects on reward processing, we agree with the reviewer that amisulpride leads to a steeper slope in the relationship between reward magnitudes and the drift rate. This means that for larger differences between reward magnitudes, amisulpride speeds up evidence accumulation towards larger-later choices (which is consistent with presynaptic effects), whereas for smaller differences between reward magnitudes amisulpride promotes choices of SS rewards (as the drift rate becomes more negative under amisulpride compared with placebo). We believe that the latter effect is difficult to reconcile with presynaptic effects, as it is commonly assumed that higher dopamine levels (reflecting presumed presynaptic effects) should strengthen the preference for larger rewards. However, the opposite seems to be true for small reward differences: the more negative drift rate under amisulpride suggests that for such offers amisulpride increases the preferences for smaller-sooner over larger-later rewards. An alternative interpretation of the result pattern therefore is that under amisulpride participants more clearly distinguish between rewards that are worth the waiting costs (large differences between reward magnitudes) and rewards that are not worth to tolerate the same waiting costs (if the larger (later) reward is only minimally larger than the smaller (sooner) reward). This in turn is consistent with the idea of a stricter cost control according to the account of Beeler and Mourra (2018), speaking in favor of reduced dopaminergic activity as results of postsynaptic effects. In the revised manuscript, we explain this reasoning in more detail, but we emphasize that for larger differences in reward magnitudes the effect could in principle also be explained by presynaptic effects (p.20-21):

“At first glance, the stronger influence of differences in reward magnitude on drift rates under amisulpride compared with placebo might therefore speak in favor of presynaptic (higher dopamine levels) rather than postsynaptic mechanisms of action in the current study. However, amisulpride versus placebo increased evidence accumulation towards LL rewards (more positive drift rate) only for larger differences between larger (later) and smaller (sooner) rewards, whereas for smaller reward differences amisulpride enhanced evidence accumulation towards SS choices (more negative drift rate; see Figure 2C). The latter finding appears inconsistent with presynaptic effects, as higher dopamine levels are thought to increase the preference for costly larger rewards (Webber et al., 2020). Instead, the stronger influence of reward differences on drift rates under amisulpride could be explained by a stricter cost control (Beeler and Mourra, 2018). In this interpretation, individuals more strongly distinguish between larger rewards that are worth the waiting costs (large difference between LL and SS rewards) and larger rewards that are not worth the same waiting costs (small difference between LL and SS rewards). While this speaks in favor of postsynaptic effects, we acknowledge that the amisulpride effects for larger reward differences are compatible with presynaptic mechanisms.”

Regarding amisulpride’s influence on the starting bias, the reviewer notes that some aspects of the data are difficult to explain via presynaptic effects: “it is unclear why presynaptic amisulpride binding would strengthen the proximity bias mostly for smaller differences in delay”. In principle though, presynaptic mechanisms could be reconciled with the current results under the additional assumption that proximity effects (at least in the current paradigm) are stronger for smaller than for larger differences in delay. We agree with this view and now clarify in the manuscript that also presynaptic effects might explain the result pattern under certain assumptions. Moreover, we re-formulated our previous statement that “amisulpride weakens the proximity advantage of SS over LL rewards”. The discussion of amisulpride effects on the starting bias now reads as follows (p.21):

“The result pattern for the starting bias parameter, in turn, suggests the presence of two distinct response biases, reflected by the intercept and the delay-dependent slope of the bias parameter (see Figure 2K), which are both under dopaminergic control but in opposite directions. First, participants seem to have a general bias towards the LL option in the current task (intercept), which is reduced under amisulpride compared with placebo, consistent with the assumption that dopamine strengthens the preference for larger rewards (Beeler and Mourra, 2018; Salamone and Correa, 2012; Schultz, 2015). Second, amisulpride reduced the impact of increasing differences in delay on the starting bias, as predicted by the proximity account of tonic dopamine (Westbrook and Frank, 2018). Both of these effects are compatible with postsynaptic effects of amisulpride. However, we note that in principle one might make the assumption that proximity effects are stronger for smaller than for larger differences in delay, and under this assumption the results would be consistent with presynaptic effects.”